# 20S proteasomes secreted by the malaria parasite promote its growth

Elya Dekel[1,13], Dana Yaffe[1,13], Irit Rosenhek-Goldian[2], Gili Ben-Nissan[1], Yifat Ofir-Birin [1], Mattia I. Morandi[1], Tamar Ziv[3], Xavier Sisquella[4,5], Matthew A. Pimentel[4,5], Thomas Nebl[4,5], Eugene Kapp[4,5], Yael Ohana Daniel[1], Paula Abou Karam[1], Daniel Alfandari[1], Ron Rotkopf [6], Shimrit Malihi[1], Tal Block Temin[1], Debakshi Mullick[6], Or-Yam Revach[1], Ariel Rudik[1], Nir S. Gov [7], Ido Azuri [6], Ziv Porat [8], Giulia Bergamaschi[9], Raya Sorkin[10,11], Gijs J. L. Wuite[9], Ori Avinoam [1], Teresa G. Carvalho [12], Sidney R. Cohen [2], Michal Sharon [1✉] & Neta Regev-Rudzki [1✉]

Mature red blood cells (RBCs) lack internal organelles and canonical defense mechanisms, making them both a fascinating host cell, in general, and an intriguing choice for the deadly malaria parasite *Plasmodium falciparum* (*Pf*), in particular. *Pf*, while growing inside its natural host, the human RBC, secretes multipurpose extracellular vesicles (EVs), yet their influence on this essential host cell remains unknown. Here we demonstrate that *Pf* parasites, cultured in fresh human donor blood, secrete within such EVs assembled and functional 20S proteasome complexes (EV-20S). The EV-20S proteasomes modulate the mechanical properties of naïve human RBCs by remodeling their cytoskeletal network. Furthermore, we identify four degradation targets of the secreted 20S proteasome, the phosphorylated cytoskeletal proteins β-adducin, ankyrin-1, dematin and Epb4.1. Overall, our findings reveal a previously unknown 20S proteasome secretion mechanism employed by the human malaria parasite, which primes RBCs for parasite invasion by altering membrane stiffness, to facilitate malaria parasite growth.

[1] Department of Biomolecular Sciences, Weizmann Institute of Science, Rehovot 7610001, Israel. [2] Department of Chemical Research Support, Weizmann Institute of Science, Rehovot 7610001, Israel. [3] Smoler Proteomics Center, Department of Biology, Technion - Israel Institute of Technology, Haifa, Israel. [4] The Walter and Eliza Hall Institute of Medical Research, 1G Royal Parade, Parkville, VIC 3052, Australia. [5] Department of Medical Biology, The University of Melbourne, Grattan Street, Parkville, VIC 3010, Australia. [6] Bioinformatics Unit, Department of Life Sciences Core Facilities, Weizmann Institute of Science, Rehovot 7610001, Israel. [7] Department of Chemical and Biological Physics, Weizmann Institute of Science, Rehovot 7610001, Israel. [8] Flow Cytometry Unit, Life Sciences Core Facilities, Weizmann Institute of Science, Rehovot 7610001, Israel. [9] Department of Physics and Astronomy and LaserLab, Vrije Universiteit Amsterdam, Amsterdam, the Netherlands. [10] School of Chemistry, Tel Aviv University, Tel Aviv, Israel. [11] Center for Physics and Chemistry of Living Systems, Tel Aviv University, Tel Aviv, Israel. [12] Department of Physiology, Anatomy and Microbiology, La Trobe University, Melbourne, VIC 3086, Australia. [13] These authors contributed equally: Elya Dekel, Dana Yaffe. ✉email: michal.sharon@weizmann.ac.il; neta.regev-rudzki@weizmann.ac.il

Red blood cells (RBCs) are endowed with exceptional features. During their ~120-day lifespan, they travel in circulation a total distance of 500 km[1], but they have already lost their internal organelles[2,3]. Throughout their maturation process, RBCs undergo drastic remodeling as they extrude their nucleus, mitochondria, Golgi apparatus, endoplasmic reticulum, and 20–30% of their cell-surface and protein-synthesis machinery[2,3]. This means that the survival of these fundamental cells relies exclusively on their internal pool of proteins[4].

The primary function of RBCs is to provide oxygen to all the body's tissues. This is enabled by their enhanced ability to alter their shape (deformability), allowing them to pass through the smallest capillaries of the human body[5]. The shear elastic properties of the RBCs are predominantly determined by the cytoskeleton's network of integral membrane proteins. The dynamic remodeling ability of the cytoskeleton network allows the repeated large deformation of RBCs, which facilitates movement[6]. Thus, their unconventional cell composition (predominantly proteins) and remarkable mechanical properties make them unique host cells for pathogens[7,8].

A case in point is the malaria parasite *Plasmodium falciparum* (*Pf*), one of the most virulent malaria species, which only infects humans and is responsible for the greatest number of fatalities worldwide, mainly among children and pregnant women[9]. The *Pf* parasites in the merozoite stage of their blood cycle (asexual replication) must invade naïve RBCs every 48 h to sustain a blood-stage infection: Once an extracellular merozoite invades the RBC, it progresses to the ring stage, grows into a metabolically active trophozoite, and, following nuclear divisions, matures into a multi-nucleated schizont[4,9]. Subsequently, new merozoites egress from each schizont and invade naïve RBCs to repeat the blood cycle. Thus, malaria pathogenesis highly depends on the extensive remodeling of host RBCs[10]. In particular, these parasites modulate RBC's membrane stiffness[6,7] and activate a phosphorylation cascade that includes RBC cytoskeletal proteins[6].

It is well known that pathogens remodel the host membrane to mediate their invasion by activating communication mechanisms that target the host cell[11,12]. However, it is still unknown whether malaria parasites, while growing inside their RBC host, remodel naïve surrounding RBCs in preparation for invasion. One known communication mechanism that pathogens employ to persist within their host is the release of multipurpose extracellular vesicles (EVs)[13,14]. These secreted vesicles are heterogeneous not only in size (ranging between 30 and 500 nm in diameter), but also in content, cell of origin and target destination (reviewed in refs. [15,16]). Their diverse cargo includes proteins, lipids, glycans, RNA, and DNA, which facilitate changes in target host cells upon internalization of the pathogenic EVs (reviewed in refs. [16,17]). We and others have previously shown that malaria parasites release EVs during their blood-stage to deliver cargo between them and to their host[18–23]. Specifically, the parasites deliver cargo to immune and endothelial host cells to promote their growth[6,19,20]. Yet, so far, it remains elusive whether the secreted vesicles are designed to modify their direct and most essential host cells, the RBCs.

To uncover the physiological role of *Pf*-derived EVs in RBC hosts, we purified vesicles from malaria infected (i) RBCs and introduced them to naïve RBCs. Remarkably, we found that the EV treatment significantly improves parasite growth. Since these host cells lost their capacity to synthesize proteins, we focused on the EV protein cargo and showed that the *Pf*-derived EVs are enriched with parasitic and host kinases as well as proteasome subunits. Strikingly, we demonstrated, using biochemical methods, that intact and functional 20S proteasome complexes are encapsulated within the vesicles (EV-20S) and it is their function that promotes parasite invasion in the treated naïve RBCs.

Further systematic mechanical analysis of the recipient RBCs revealed that EV-20S proteasome activity increases cell membrane deformability and disturbs the structural integrity of the host cytoskeleton. Moreover, treatment with *Pf*-derived EVs prompted distinct phosphorylation events within unstructured regions of four cytoskeletal proteins, all of which belong to the same network machinery of the RBC cell cortex: β-adducin, ankyrin-1, dematin, and erythrocyte membrane protein band 4.1 (Epb4.1). Finally, we show that these phosphorylated proteins serve as a direct target for degradation by the delivered EV-20S proteasome. Taken together, our data indicate that assembled and functional 20S proteasome complexes secreted from malaria-infected RBCs are able to prime naïve host RBCs for impending parasite invasion. Overall, this study reveals a mechanism in which intracellular parasites utilize EVs in order to reshape their essential host cells.

## Results

### EVs derived from *Pf*-iRBCs promote parasite growth in human RBCs.

To examine whether *Pf*-derived EVs affect naïve RBCs, we purified EVs from RBCs infected (i) with *Pf* NF54 strain[19] (Supplementary Fig. S1A–C). The *Pf* parasite is constantly cultured in human RBCs pooled from healthy donors from which *Pf*-derived EVs are isolated. As control, we extracted vesicles harvested from uninfected (u) RBCs. Naïve RBCs were pre-incubated with equal amounts of EVs derived from uRBCs or ones derived from *Pf*-iRBCs (EV levels were counted by NTA measurement, Supplementary Fig. S1C). Magnet-purified *Pf* parasites were then fed by the two groups of pre-treated RBCs and parasitemia levels were monitored for two blood cycles (four days) by both counting Giemsa smears of ring-stage parasites and by a FACS-based assay[24] (Fig. 1A, B and Supplementary Fig. S2). Our results indicate that in the naïve RBC group that received the *Pf*-derived EV pre-treatment, parasitemia levels increased by ~29% in comparison to those pre-treated with the control EVs (Fig. 1A). This finding suggests that EVs secreted from parasitic-infected RBCs alter naïve host RBCs to favor parasite growth.

### *Pf*-derived EVs alter human RBC stiffness by disrupting the cytoskeleton network.

It was previously shown that increased deformability of the naïve RBC membrane is required for parasite invasion and successful growth[6]. Therefore, we applied an Atomic Force Microscopy (AFM)-based approach to examine whether *Pf*-derived EVs alter the mechanical properties of naïve RBCs. In evaluating this test we treat the composite thin shell consisting of membrane bilayer and the adsorbed cortical cytoskeleton as a single unit which presumably bears responsibility for the mechanical response of the cell, as opposed to the bulk of the cell volume comprising cargo and fluids. EVs harvested from iRBC or uRBC cultures were incubated with human RBCs that were pooled from a total of six healthy donors. The RBCs treated with *Pf*-derived EVs expressed significantly lower effective Young's modulus values (Fig. 1C, blue bars), while control EVs did not significantly alter the host cell membrane's properties (Fig. 1C, red bars). Specifically, Young's modulus was reduced from a median value of 461 Pa within a broad distribution for the untreated control to a much tighter distribution with median of 148 Pa for the *Pf*-derived EV-treated RBCs. A Tukey multiple comparison of the means yielded a non-significant difference ($p = 0.074$) between cells exposed to uRBC-derived EVs and those unexposed to EVs but very significant difference ($p < 0.0001$) between both of those two controls and the *Pf*-derived EV-treated cells. To further validate the mechanical effect of *Pf*-derived EVs on RBC mechanics, we have performed cell-pulling measurements using acoustic force spectroscopy (AFS)[25], using

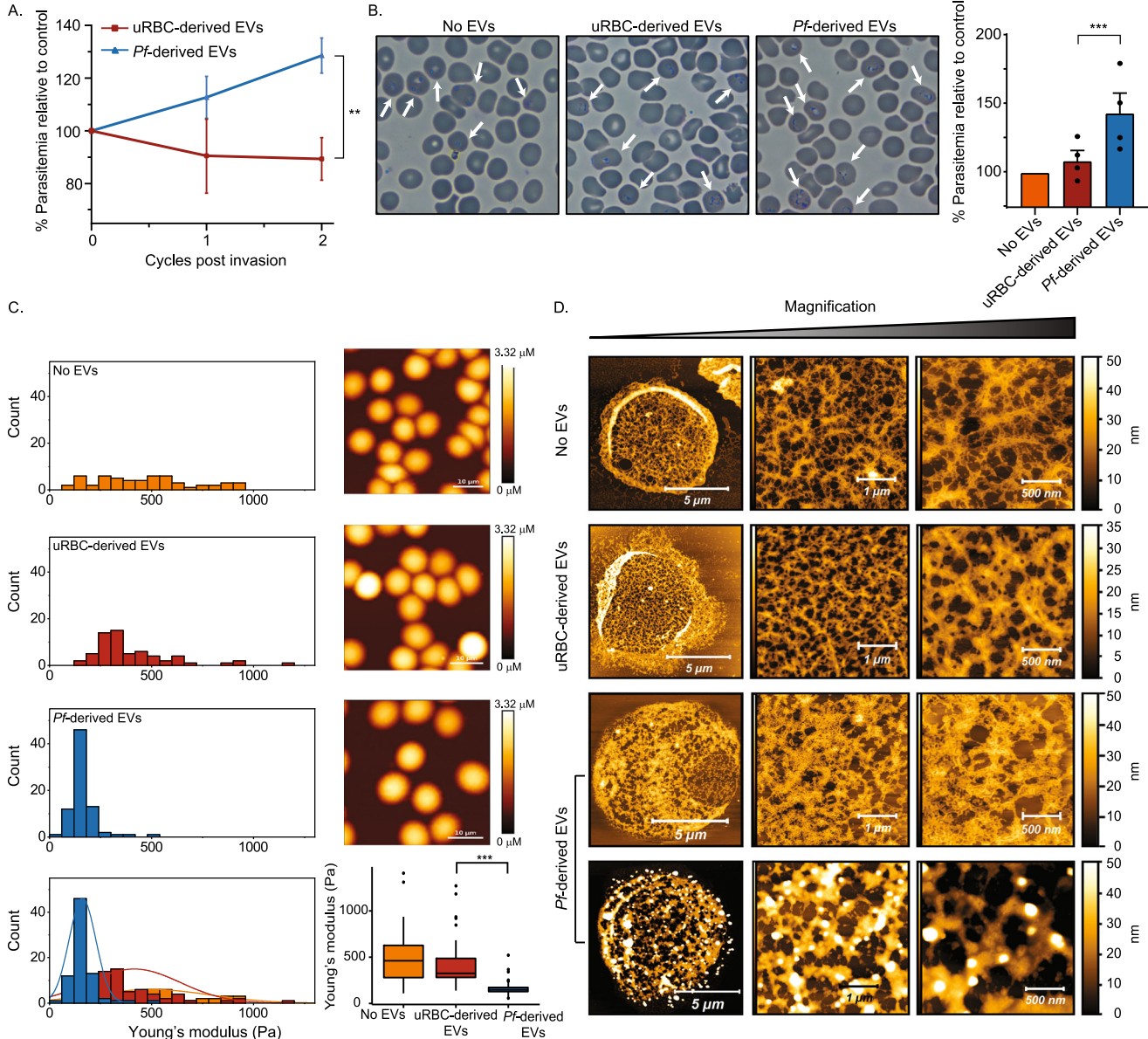

**Fig. 1 *Pf*-derived EV treatment influences growth dynamics and morphology of naïve RBCs. A, B** *Pf*-derived EVs were incubated for 18 hr with naïve RBCs. Magnet-purified *Pf* were then introduced to the treated cells and the parasitemia level was monitored (estimated comparing to the untreated control) using flow cytometry (**A**) and Giemsa smears (**B**). Treatment with uRBC-derived EVs or the absence of EVs were used as controls. **A** Averages and statistical significance for at least three independent experiments were calculated by comparing percentages (relative to control) between treatments with a 2-way ANOVA, accounting for treatment and batch (uRBC-derived EVs 1st cycle $n = 3$, *Pf*-derived EVs 1st cycle $n = 9$, uRBC-derived EVs 2nd cycle $n = 4$, *Pf*-derived EVs 2nd cycle $n = 12$, **$p = 0.00416$). Error bars represent SEM (standard error of the mean). **B** The arrows indicate *Pf*-iRBCs. Scale bars represent 10 μm. Averages and statistical analysis of four independent experiments, in which counts were compared between treatments using a mixed GLM, assuming a binomial distribution, with a random intercept for batch (***$p = 0.0008$). Error bars represent SEM. **C** Mechanical changes of the recipient naïve RBCs following the *Pf*-derived EV treatment. Distributions of Young's modulus values were obtained from AFM measurements (left panel), each count in the histogram represents the average value calculated from 25 indentations near the center of each cell. AFM images of the EV-treated and -untreated RBCs are shown on the right panel (scale bar 10 μm). Differences between groups shown in the box plot were tested with a one-way ANOVA, followed by Tukey's post-hoc test. Boxes represent the 25-75 percentiles of the sample distribution, with black vertical lines representing the 1.5×IQR (interquartile range). The black dots represent outliers. Black horizontal line represents the median. Log-transformed data was used due to the mean-variance correlation. Three independent experiments, each comparing naïve RBCs, uRBC-derived EV and *Pf*-derived EV exposure, were performed with total of 15 full images acquired for each of the cell exposure conditions. Number of cells measured were for No EVs $n = 57$, uRBC-derived EVs $n = 63$, *Pf*-derived EVs $n = 77$, ***$p < 0.001$). **D** Representative AFM images of cytoskeletal structures of naïve RBCs, naïve RBCs incubated with either uRBC-derived or *Pf*-derived EVs. The sharp fibrilar network assigned to spectrin filaments seen in the upper frames breaks down and becomes blurred and disconnected in the lower frames. Three independent experiments were performed, with a total of 50 AFM scans for each type of cell treatment. **A–D** Results are representative of at least three independent biological replicates. Source data for A and B are provided in Supplementary Fig. S2.

an experimental approach as described in detail in Sorkin et al.[26]. In our approach, an acoustic pressure field is used to apply well-controlled forces up to 500 pN to cells confined between a microsphere and a surface. The microspheres which are attached on top of cells are pushed towards the nodes of the acoustic pressure field, thereby pulling on the cells. We track the position of the beads in three dimensions using a standard optical microscope, and thus can monitor the elongation of cells under applied force. Using this approach, we found that RBCs treated by *Pf*-derived EVs elongated significantly more than non-treated RBCs (Supplementary Fig. S3). Following calibration of the applied forces, the spring constants are calculated for each cell (that is, force divided by elongation), as shown in Supplementary Fig. S3. Lower spring constants are obtained for the EV-treated RBCs, in line with lower Young's modulus measured by AFM.

The mechanical properties of the RBCs are strongly influenced by the underlying cytoskeleton network[7,8,27]. We therefore used high-resolution AFM imaging to assess whether the morphology of the RBC cytoskeleton network was affected by the presence of *Pf*-derived EVs[7]. As seen by the representative phenotypes in Fig. 1D, for the healthy RBC, the cytoskeleton comprises a filamental network. These features have been assigned to spectrin filaments and were also observed in previous AFM studies of the RBC (see, for example refs. [7,28,29]). The filaments are known to be connected to the membrane at protein complexes that serve as the nodes of this two-dimensional, membrane-bound network. In the *Pf*-derived EV-exposed human RBCs, the network structure is disrupted: in some cells we find regions of the membrane where the node-filament network seems to be dissolved and replaced by a more uniform distribution of material (top row of the *Pf*-derived EV-exposed cells in Fig. 1D). In other cells, large-scale aggregation into mounds of material is evident, which indicates that the original network is highly disrupted (bottom row of the *Pf*-derived EV-exposed cells in Fig. 1D). This modulation of the cytoskeleton was not observed when naïve RBCs were treated with control vesicles derived from healthy RBCs (Fig. 1D), indicating that only the parasitic vesicles induced these cytoskeleton alterations in naïve RBCs.

**The *Pf*-derived EV cargo is enriched with kinases and 20S proteasome subunits**. Considering that mature RBCs lack translational machinery, it is reasonable to speculate that the observed *Pf*-derived EV-induced cytoskeleton changes are mediated by the delivered protein cargo (rather than the nucleic acid cargo). Therefore, we tested whether EVs derived from *Pf*-iRBCs are able to fuse with target membranes. To this end, we used a FRET-based fusion assay[30–32], between *Pf*-derived EVs and liposomes resembling the plasma membrane lipid composition of the RBC[33]. *Pf*-derived EVs display significant membrane mixing compared to large unilamellar vesicles (LUVs) (Supplementary Fig. S4A), suggesting EVs can fuse with the RBC membrane to deliver their content. Moreover, using nanoparticle tracking analysis (NTA) we measured the size distribution of the vesicles before and after mixing of EVs with LUVs and found that vesicle size increases after mixing, due to the addition of membrane (Supplementary Fig. S4B–D), most probably via fusion. Taken together, these results suggest that EVs fuse to deliver their cargo.

Next we employed proteomics analyses to reveal the protein cargo carried by *Pf*-derived EVs. EVs were harvested from *Pf*-iRBC culture and their proteins were subsequently extracted and subjected to LC-MS/MS analysis. A total of 800 human proteins were identified (Supplementary Table 1A), in addition to 387 *Pf* proteins (Supplementary Table 1B). Among the human host proteins were 304 cellular components of extracellular exosomes (GO:0070062, 2.7-fold enrichment, *p* value 3.30E-67, FDR 8.84E-67).

These included SR1, ENO1, ALDOA, HSP90AA1, GAPDH, HSPA8, PGK1, ACTB, CFL1, TF, YWHAZ, LDHA, YWHAE, CLTC, RAB5A, PDCD6IP, all markers typically found in human EVs[34–36] (https://www.antibodies-online.com/images/resources/ABO_Exosome_poster.pdf). Interestingly, Gene Ontology (GO) analysis based on cellular component enrichment revealed that the most enriched term in *Pf*-derived EVs represents a family of parasitic kinases (Fig. 2A, B and Supplementary Table 1B). We also noticed that proteasome subunits, of both human and *Pf* origin, are among the top-ranking components (Fig. 2A, C, D, F and Supplementary Table 1A, B).

The proteasome is a conserved degradation machinery essential for maintaining cellular homeostasis[37]. It comprises two functional species, the 26S and 20S proteasome complexes, which are not mutually exclusive[38,39]. Degradation by the 26S proteasome, which is composed of the 19S and 20S particles, is an active process that is regulated by a series of enzymes that ubiquitinate the substrate and sensitize it to degradation[40]. In contrast, the 20S proteasome, on its own, can degrade protein substrates in a ubiquitin- and ATP-independent manner, by recognizing unfolded or unstructured regions within its substrates[38,41]. Hence, in order to validate the presence of 20S and/or 26S proteasome subunits within the *Pf*-derived vesicles, we purified the EVs by sedimentation into a density gradient via an OptiPrep centrifugation assay[34]. Western blot analysis of the EV pool (fractions 6–8) confirmed the presence of the 20S proteasome subunits PSMA1 (Fig. 3A), with SR1 and HSP90 serving as EV control markers. We were also able to identify PSMD1, a 19S subunit which forms the 26S proteasome (Fig. 3A).

We validated that protein kinases within the secreted *Pf*-derived EVs are active, by monitoring Ser/Thr phosphorylation events that were increased upon ATP addition to isolated and sonicated *Pf*-derived EVs (Supplementary Fig. S5). Overall, our results imply that *Pf*-derived EVs contain kinases and 20S proteasome subunits, suggesting that these delivered proteins play a role in the observed host mechanical alterations and the resultant elevated parasitemia levels.

**Pf-derived EVs affect phosphorylation of the RBC cytoskeleton**. To investigate whether the delivered kinases (Fig. 2B, E) are involved in phosphorylation events of the host human RBCs, we treated naïve RBCs with *Pf*-derived EVs for either 5 or 15 min and then subjected them to phosphoproteomics analyses. In order to eliminate background noise from hemoglobin and cytosolic proteins, we focused our analysis on the ghost RBC fraction (i.e., the membrane component and bound cytoskeletal proteins). Treatment with EVs derived from healthy naïve RBCs served as control. In total, 930 distinct phosphorylation sites from 394 proteins were identified for each condition in at least 2 samples out of the 6 biological repeats (Supplementary Data 1). As samples from different blood donors give rise to different phosphorylation kinetics and different responses, we focused on phosphosites that were unique to the *Pf*-derived EVs. Namely, those that were identified in at least 3 out of the 6 samples in either time point and differential intensities with at least 2-fold change (*p* value < 0.1), or those that were identified in at least 3 *Pf*-derived EV samples and in none of the uRBC (Table 1). A total of 55 phosphosites corresponding to 43 proteins were elevated upon treatment with parasitic EVs, of which, 20 were unique to the parasitic EVs. Importantly, GO Biological Processes analysis of the differential phosphosites shows a clear enrichment of localization (FDR 0.00029) and transport (FDR 0.00074) proteins as well as those involved in cytoskeleton organization (FDR 2.24E-05) and actin cytoskeleton organization (FDR 4.03E-05)

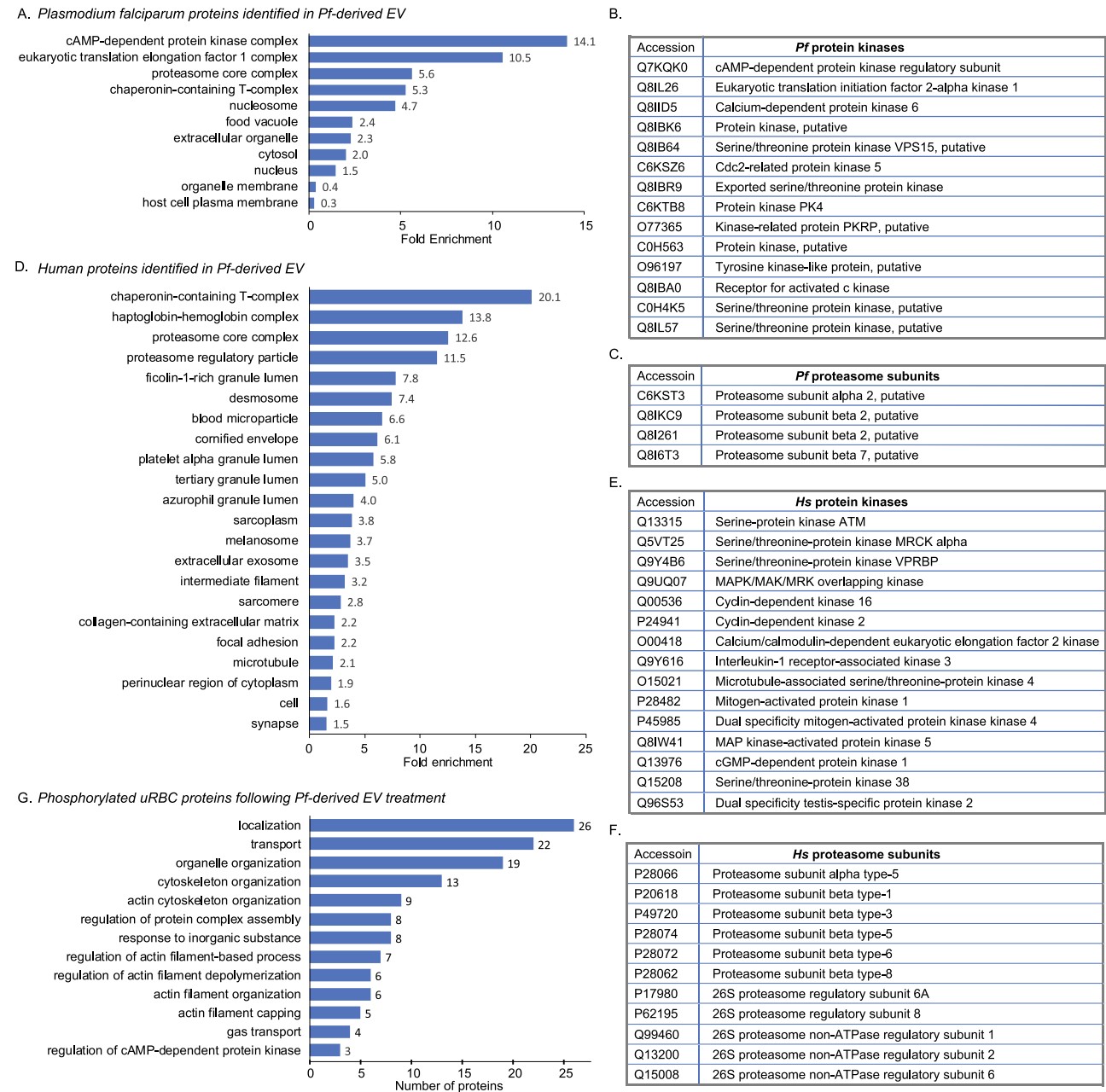

**Fig. 2 Pf-derived EVs are enriched with kinases and proteasome subunits.** Gene Ontology (GO) cellular components enrichment analysis of *Pf*-derived EV protein cargo. Vesicles were harvested and proteins were extracted and subjected to cellular components analysis following LC-MS/MS proteomic identification. The horizontal bar graph shows the fold enrichment of significantly enriched **A** *Pf* and **D** human proteins. Results are based on GO cellular component enrichment analysis of all identified proteins against the background of **D** *Homo sapiens* and **A** *Plasmodium falciparum* (http://geneontology.org/; *p*-value < 0.05, FDR < 0.01). Lists of identified *Pf* and human (**B** and **E**) kinases and proteasome subunits (**C** and **F**). **G** Phosphoproteomics analysis of uRBCs treated with *Pf*-derived EVs. Identified proteins were subjected to GO Biological Processes analysis (FDR < 0.00035). Statistical analyses in **A** and **D** were done using PANTHER Statistical Overrepresentation Test (Released 2019-07-11). Annotation versions and release date: GO Ontology database (Released 2019-10-08), PANTHER version 15 (Released 2020-02-14). Analyzed lists: *Homo sapiens* or *Plasmodium falciparum* Proteins IDs (Uniprot Accessions are listed in Table S1A, S1B). Reference list: *Homo sapiens* or *Plasmodium falciparum* (all genes in database). Test Type: FISHER, applying FDR correction, *p* < 0.05.

(Fig. 2G). Eight of the 11 cytoskeleton organization proteins comprising the unique phosphosites belong to the same protein cytoskeletal complex, namely, α-adducin (ADD1), β-adducin (ADD2), γ-adducin (ADD3), ankyrin-1 (ANK1), dematin (DMTN), Ebp41, spectrin α−chain (SPTA1) and spectrin β−chain (SPTB). These findings could underlie the AFM results demonstrating significant alteration of both the stiffness and morphology of the cytoskeleton.

To further validate the phosphoproteomics results, we immunoprecipitated dematin (one of the modified cytoskeleton proteins) from human RBCs treated with *Pf*-derived EVs (Supplementary Fig. S6) and performed an additional phospho-proteomics analysis on the pulled down fraction. We chose dematin due to the multiple *Pf*-derived EV-dependent phosphor-ylation sites identified on it (Supplementary Data 1). Again, dematin was phosphorylated at position Ser333 following

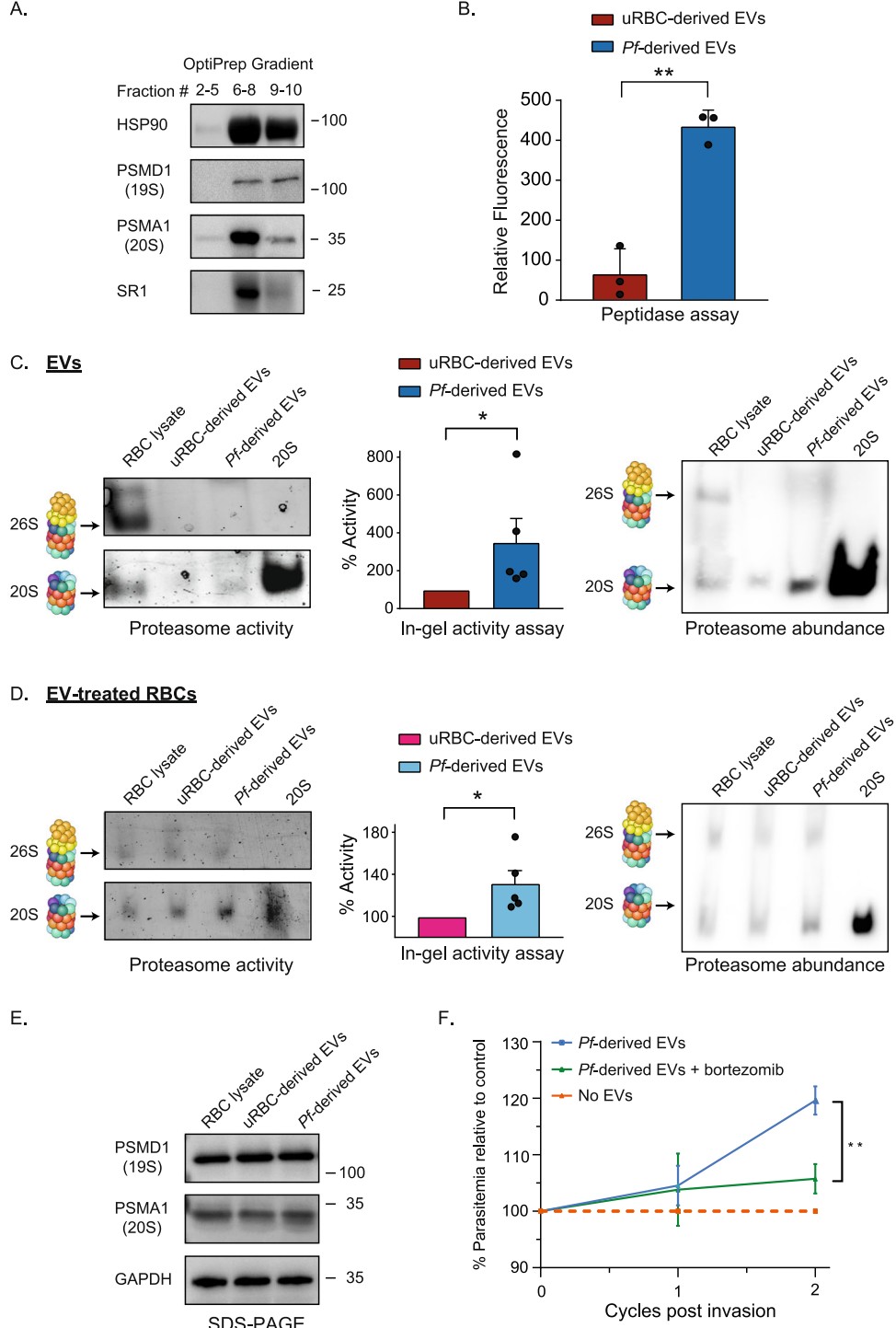

*Pf*-derived EV treatment of naïve human RBC (Supplementary Fig. S6B, C). These results indicate that *Pf*-derived EVs induce a distinct phosphorylation pattern in cytoskeleton proteins of the RBC host.

Notably, we also found a unique phosphorylation on glycophorin-A (GYPA) after the parasitic EV treatment (Table 1). GYPA is known to be a receptor for *Pf* erythrocyte-binding antigen 175 (EBA-175)[6]. Thus, the phosphorylation can be part of the parasitic regulation. In addition, unique phosphorylations were seen on several channels: calcium-transporting ATPase-ATP2B4, SLC2A1-glucose transporter, AQP1 water-specific channel, and PIEZO1, a mechanically-activated ion channel.

PIEZO1 was previously identified to be phosphorylated in other sites following *Pf* invasion[42].

**An intact, assembled and functional 20S proteasome is delivered by *Pf*-derived EVs to naïve RBCs.** Given the large number of proteasome subunits identified within *Pf*-derived EVs (Fig. 2), we questioned whether their presence is linked to the membrane mechanical alterations and, thus, to the EV-mediated enhancement of parasite growth (Fig. 1A). We therefore examined whether the EV-20S subunits (Fig. 3A) are not just present but actually assembled and functional within the secreted vesicles derived from *Pf*-iRBCs. Initially, the peptidase activity of the

**Fig. 3 The 20S proteasome is assembled and functional within *Pf*-derived EVs. A** Western blot analysis of the different fractions separated by OptiPrep velocity gradient centrifugation. Anti-PSMA1 antibody was used to probe the 20S proteasome complex, and anti-PSMD1 was used to identify the 19S particle. As a positive control, antibodies against the EV markers HSP90 and SR1 were used. This experiment was repeated independently three times with similar results. **B** Proteasome proteolytic activity measurement of *Pf*-derived EVs with the fluorescent peptide Suc-LLVY-AMC. As a control, uRBC-derived EVs were used. The data was normalized with respect to background levels of fluorescence in the presence of the proteasome inhibitor MG132. Averaging and statistical analysis were performed on three biological replicates using two tailed *t*-test assuming unequal variances. Error bars represent SD (standard deviation) (\*\*$p = 0.0032$). **C** To validate the integrity and activity of the proteasome complex, uRBC-derived, and *Pf*-derived EVs were lysed and separated using 4% native-PAGE. The activity of the proteasome complexes was analyzed by incubating the gels with the fluorescent peptide substrate Suc-LLVY-AMC (left panel). Quantifications demonstrates the average of five independent biological replicates subjected to paired two-way *t*-test analysis using log-transformed measurements (\*$p = 0.028$). Error bars represent SEM. The relative abundance of the complexes was assessed by Western blot analysis of the native gel (right panel, showing a representative gel from three independently repeated experiments with similar results.) using an anti-PSMA1 antibody. Samples of RBC lysate and purified 20S proteasome were used as controls. **D** Naïve RBCs were incubated with EVs purified from *Pf*-iRBCs or from uRBC cultures; samples were then lysed and separated using 4% native-PAGE and incubated with the fluorescent peptide (left panel) for activity analysis. Band intensity quantifications of five independent experiments were subjected to averaging and paired t-test analysis (\*$p = 0.021$). Error bars represent SEM. Western blot of the native gel (right panel), using an anti-PSMA1 antibody. Samples of uRBCs and uRBCs treated with EVs derived from naïve RBCs were used as controls. **E** Denaturing gel of the samples analyzed in **D**. Anti-PSMA1 was used to probe the 20S proteasome, and anti-PSMD1 was used to probe the 19S complex. Glyceraldehyde-3-phosphate dehydrogenase (GAPDH) was used as a loading control. All images are representatives of three independent repeats (see Supplementary Fig. S9). **F** Parasite growth curve in EV-treated RBCs following incubation with *Pf*-derived EVs with or without the proteasome inhibitor bortezomib. Average and statistical significance was calculated for at least three independent experiments by comparing percentages (relative to control) between treatments with a two-way ANOVA, accounting for treatment and batch (No EVs $n = 3$, *Pf*-derived EVs $n = 7$, *Pf*-derived EVs + bortezomib $N = 7$), \*\*$p = 0.0015$). Error bars represent SEM. Source data are provided in Supplementary Fig. S7 and in the source data file.

proteasome was monitored in *Pf*-derived EV extracts using fluorogenic proteasome substrates[43] (Fig. 3B and Supplementary Fig. S7A and S8). All three types of proteasome enzymatic activities were detected in the *Pf*-derived EVs, namely, caspase-like, chymotrypsin-like and trypsin-like. Using this assay, we found that the proteasome complex is highly active in *Pf*-derived EVs as opposed to control EVs (Fig. 3B).

Next, we examined whether the secreted EVs contain 26S and/ or 20S assembled proteasome complexes. This was achieved by separating EV extracts on native-PAGE, followed by performing an in-gel proteasome activity assay (Fig. 3C left panel and Supplementary Fig. S9). The analysis revealed that *Pf*-derived EVs exhibit enhanced peptidase activity in comparison to EVs derived from uRBCs (Fig. 3C middle panel and Supplementary Fig. S7B). Moreover, unlike uRBC lysates, which contain both active 26S and 20S proteasome species, *Pf*-derived EVs contain the active, assembled 20S proteasome complex, but not the 26S particle. Western blot analysis using an antibody against the human PSMA1 subunit validated the presence of 20S proteasome particles in the *Pf*-derived EVs (Fig. 3C, right panel and Supplementary Fig. S9), as opposed to the low abundance of the 26S proteasome. Moreover, Western blot analysis with an anti-ubiquitin antibody of RBC lysates, before and after incubation with *Pf*-derived or control EVs, and of the EVs themselves, revealed no significant change in global ubiquitination levels (Supplementary Fig. S10). Taken together, our results indicate that functional 20S (and not 26S) proteasome complexes are found within *Pf*-derived EVs, suggesting that the restricted volume in the vesicles promotes 20S-mediated degradation rather than ubiquitin-dependent proteolysis. Although we are currently unable to precisely delineate the composition of the subunits of the EV-20S proteasome complex (parasitic vs. human), to the best of our knowledge, this is the first report exhibiting an intact and functional 20S proteasome encapsulated within EVs derived from infected RBCs.

To determine whether the presence of the 20S proteasome in *Pf*-derived EVs affects the global activity of 20S proteasomes in recipient naïve RBCs, we measured the 20S proteolytic activity in naïve RBCs before and after treatment with *Pf*-derived EVs. Indeed, our results indicate a moderate increase in total 20S proteasome activity in RBCs following the *Pf*-derived EV treatment (Fig. 3D and Supplementary Figs. S7C and S9),

suggesting that the *Pf*-EV-20S proteasome increases the cellular proteolysis capacity upon uptake. This enhancement is also evident when comparing the 20S proteasome activity to that of the control experiment, in which RBCs were treated with EVs derived from uRBCs (Fig. 3D and Supplementary Figs. S7C and S9). Immunoblotting of native and denatured gels indicates no significant change in the level of both the 20S and 26S proteasomes (Fig. 3D right panel, 3E and Supplementary Fig. S9). However, this is expected, given the dilution of the EV protein content within the recipient cell and that in in vitro *Pf* cultures only ~5–8% of the total RBCs are infected by the parasites, suggesting that the specific increase in EV-derived 20S proteasome activity is relatively large.

Importantly, when we repeated the parasite growth experiment, but this time in the presence or absence of bortezomib, which reversibly inhibits the chymotrypsin-like activity of the proteasome[44], we detected a significant decrease in parasite growth in the presence of the *Pf*-derived EVs and the proteasome inhibitor as opposed to treatment with *Pf*-derived EVs alone (Fig. 3F and Supplementary Fig. S7D). As a control, we validated that this observation is not due to the presence of bortezomib, by comparing parasitemia levels of RBC in the presence or absence of the proteasome inhibitor after pre-incubation with control EVs derived from uRBCs (Supplementary Fig. S11). Taken together, these findings suggest that the encapsulated 20S proteasome complexes play a role in promoting parasite invasion and thus its growth.

### The secreted EV-20S proteasome alters the mechanical properties of human RBCs and disrupts the cytoskeleton network.

To test whether the observed modulation in RBC stiffness upon *Pf*-derived EVs treatment (Fig. 1C, D) is associated with proteasome activity, we treated prior to incubation with naïve human RBC, *Pf*-derived EVs with proteasome inhibitors and measured, by way of AFM, the stiffness of the cells (Fig. 4 and Supplementary Fig. S12). As *Pf* parasites are growing in RBCs pooled from healthy donors, we replicated the mechanical measurements in three independent biological repeats of parasite cultures, which were introduced in each repeat to minimum of two different healthy donors (Fig. 4A and Supplementary Fig. S12). Corroborating our previous results (Fig. 1C), our findings show that the

**Table 1 Phosphosites that were enriched following Pf-derived EV treatment.**

| Phosphosite | | | | No. of identifications | | | | |
|---|---|---|---|---|---|---|---|---|
| Protein | Protein name | Position | Gene names | *Pf* 5 min | *Pf* 15 min | uRBC 5 min | uRBC 15 min | Amino acid |
| P35611 | α-adducin | 364 | ADD1 | 3 | 5 | 1 | 1 | T |
| P35612 | β-adducin | 699 | ADD2 | 4 | 2 | 0 | 2 | S |
| Q9UEY8 | γ-adducin | 679 | ADD3 | 3 | 1 | 0 | 0 | S |
| P16157 | Ankyrin-1 | 1075 | ANK1 | 3 | 2 | 0 | 1 | T |
| P16157 | Ankyrin-1 | 960 | ANK1 | 6 | 6 | 3 | 3 | S |
| P16157 | Ankyrin-1 | 1696 | ANK1 | 5 | 6 | 2 | 3 | S |
| P16157 | Ankyrin-1 | 1646 | ANK1 | 1 | 2 | 0 | 0 | S |
| H7BZV9 | Ankyrin repeat domain 54 | 76 | ANKRD54 | 3 | 1 | 0 | 1 | S |
| K7N7A8, P29972 | Aquaporin-1 | 426 | AQP1 | 0 | 3 | 0 | 0 | T |
| P23634 | ATPase plasma membrane Ca2+ | 1162 | ATP2B4 | 0 | 4 | 0 | 0 | S |
| O14523 | C2CD2 like | 470 | C2CD2L | 3 | 2 | 0 | 1 | S |
| O14523 | C2CD2 like | 613 | C2CD2L | 1 | 2 | 0 | 0 | S |
| P00918 | carbonic anhydrase 2 | 165 | CA2 | 3 | 1 | 0 | 2 | S |
| E7EQ12 | Calpastatin | 81 | CAST | 1 | 2 | 0 | 0 | S |
| E9PS23 | Cofilin 1 | 24 | CFL1 | 1 | 2 | 0 | 0 | S |
| Q08495 | Dematin | 285 | DMTN | 1 | 3 | 0 | 0 | S |
| Q08495 | Dematin | 124 | DMTN | 1 | 3 | 0 | 0 | S |
| Q08495 | Dematin | 333 | DMTN | 4 | 5 | 3 | 0 | S |
| Q7L9B9 | Endonuclease/exonuclease/ phosphatase family domain containing 1 | 173 | EEPD1 | 3 | 4 | 0 | 2 | S |
| Q8N3D4 | EH domain binding protein 1 like 1 | 191 | EHBP1L1 | 2 | 1 | 0 | 0 | S |
| E7EX73 | Eukaryotic translation initiation factor 4 gamma 1 | 1046 | EIF4G1 | 2 | 1 | 0 | 0 | S |
| P11171 | Erythrocyte membrane protein band 4.1 | 200 | EPB41 | 1 | 2 | 0 | 0 | S |
| Q5T0W9 | Family with sequence similarity 83 member B | 766 | FAM83B | 5 | 4 | 0 | 2 | S |
| Q0JRZ9 | FCH domain only 2 | 488 | FCHO2 | 3 | 3 | 0 | 1 | S |
| Q0JRZ9 | FCH domain only 2 | 496 | FCHO2 | 1 | 2 | 0 | 0 | S |
| Q86UX7 | Fermitin family member 3 | 347 | FERMT3 | 1 | 2 | 0 | 0 | T |
| E7EUT5 | Glyceraldehyde-3-phosphate dehydrogenase | 109 | GAPDH | 0 | 3 | 0 | 0 | T |
| A2AB27 | Guanine nucleotide-binding protein-like 1 | 49 | GNL1 | 4 | 4 | 0 | 2 | S |
| P02724 | Glycophorin a | 133 | GYPA | 3 | 4 | 0 | 0 | T |
| P02724, A0A087WU29 | Glycophorin a | 146 | GYPA | 3 | 4 | 0 | 2 | S |
| P68871 | Hemoglobin subunit beta | 13 | HBB | 1 | 2 | 0 | 0 | T |
| P69891 | Hemoglobin subunit gamma 1 | 53 | HBG1;HBG2 | 3 | 1 | 0 | 2 | S |
| E9PHG0 | Kell blood group, metallo-endopeptidase | 28 | KEL | 3 | 5 | 0 | 3 | S |
| Q8IY33 | Mical like 2 | 143 | MICALL2 | 5 | 6 | 3 | 1 | S |
| Q00013 | Membrane palmitoylated protein 1 | 16 | MPP1 | 0 | 3 | 0 | 0 | T |
| C9J6D1 | Nucleosome assembly protein 1 like 4 | 53 | NAP1L4 | 4 | 4 | 1 | 0 | S |
| P00558 | Phosphoglycerate kinase 1 | 203 | PGK1 | 1 | 2 | 0 | 0 | S |
| Q92508 | Piezo type mechanosensitive ion channel component 1 | 1820 | PIEZO1 | 2 | 3 | 0 | 0 | S |
| F8WE65 | Peptidylprolyl isomerase a | 77 | PPIA | 2 | 1 | 0 | 0 | S |
| P10644 | Protein kinase camp-dependent type i regulatory subunit alpha | 6 | PRKAR1A | 3 | 3 | 0 | 1 | T |
| Q00577 | Purine rich element binding protein a | 183 | PURA | 1 | 2 | 0 | 0 | T |
| Q9Y4G8 | Rap guanine nucleotide exchange factor 2 | 585 | RAPGEF2 | 1 | 2 | 0 | 0 | S |
| P11166 | Solute carrier family 2 member 1 | 490 | SLC2A1 | 3 | 4 | 0 | 2 | S |
| P02549 | Spectrin alpha, erythrocytic 1 | 421 | SPTA1 | 1 | 3 | 0 | 0 | S |
| P02549 | Spectrin alpha, erythrocytic 1 | 2191 | SPTA1 | 1 | 2 | 0 | 0 | T |
| P11277 | Spectrin beta, erythrocytic | 2128 | SPTB | 3 | 3 | 0 | 1 | S |
| B8ZZJ0 | Small ubiquitin-like modifier 1 | 2 | SUMO1 | 4 | 6 | 0 | 3 | S |
| E5RJ93 | Transcription elongation factor a1 | 57 | TCEA1 | 2 | 2 | 0 | 0 | S |
| A0A087WZA9 | Transmembrane protein 120a | 167 | TMEM120A | 0 | 4 | 0 | 0 | S |
| F8VZH8 | Unc-13 homolog a | 34 | UNC13A | 3 | 4 | 2 | 0 | T |
| F8VZH8 | Unc-13 homolog a | 35 | UNC13A | 3 | 4 | 2 | 0 | T |
| F8VZH8 | Unc-13 homolog a | 33 | UNC13A | 3 | 4 | 2 | 0 | S |
| F5GWT4 | Wnk lysine deficient protein kinase 1 | 185 | WNK1 | 4 | 4 | 0 | 2 | S |
| F5GWT4 | Wnk lysine deficient protein kinase 1 | 167 | WNK1 | 3 | 0 | 0 | 1 | S |
| O43149 | Zinc finger zz-type and ef-hand domain containing 1 | 2444 | ZZEF1 | 3 | 3 | 0 | 2 | S |

The table shows phosphosites that were unique for the *Pf*-derived EVs, which were identified in at least three samples and in none of the uRBC-derived EVs, or present differential intensities with at least 2-fold change (*p* value < 0.1).

*Pf*-derived EV treatment significantly lowers the Young's modulus values (Fig. 4A), from median of 588 Pa to 247 Pa (Fig. 4A), following the trend reported above. However, exposure of *Pf*-derived EVs to the proteasome inhibitor prior to the treatment countered the deformability effect, so that the Young's modulus values were similar to those of the control set of untreated naïve RBCs with median of 479 Pa (Fig. 4A). The Tukey multiple comparison of means shows a significant difference between the control and proteasome inhibitor-treated sample relative to the *Pf*-derived EV infected RBCs (*p* < 0.001) whereas the control and

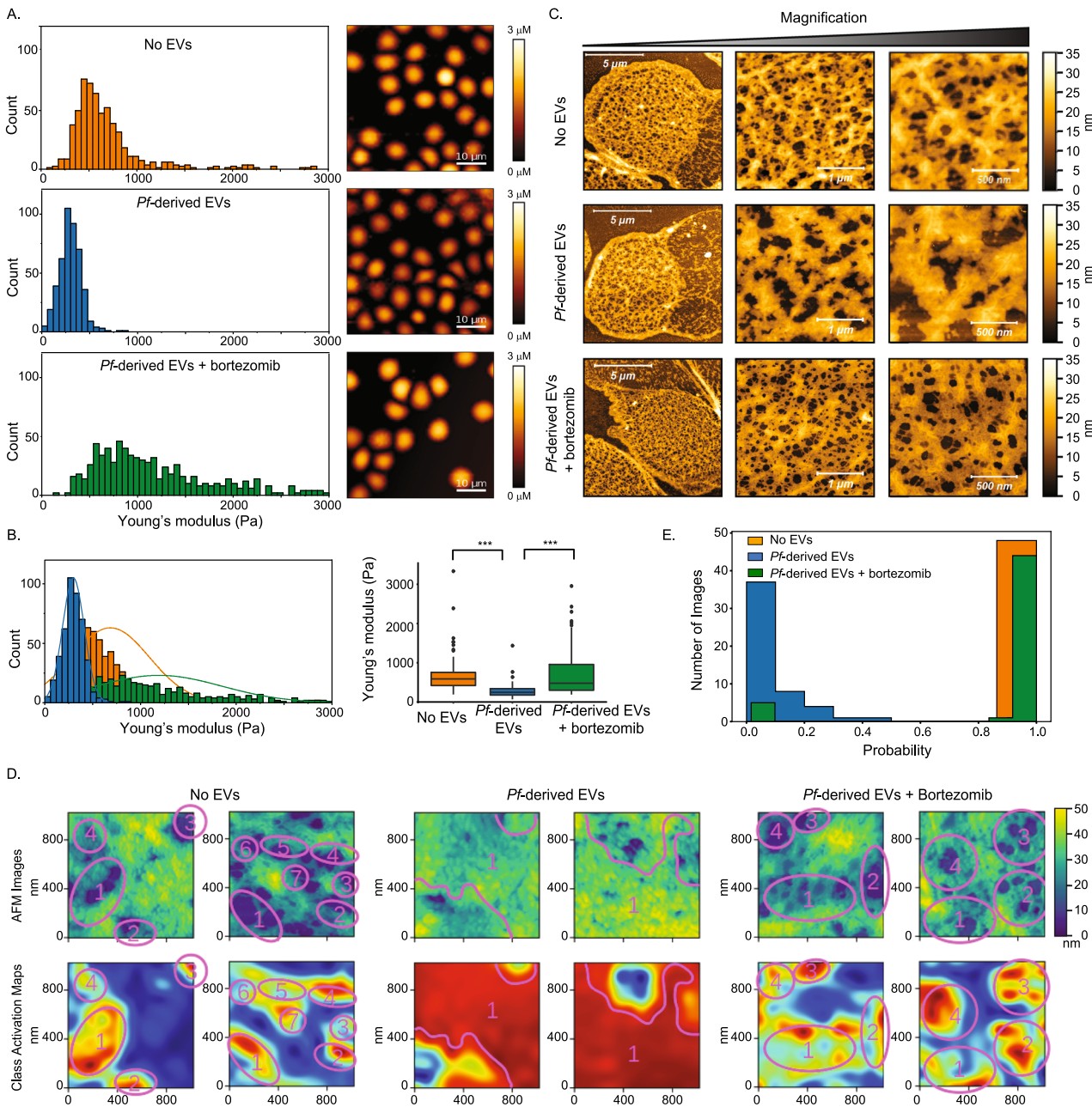

**Fig. 4 EV-20S proteasomes modulate the mechanical properties of RBCs. A** Mechanical changes measured by AFM following *Pf*-derived EV incubation with naïve RBCs. *Pf*-derived EVs or *Pf*-derived EVs pretreated with proteasome inhibitor (bortezomib) were incubated with naïve RBCs. Pretreated cells were deposited on a mica surface for AFM topography and mechanical measurements. Distributions of Young's modulus values shown in the left panel, each count in the histogram represents data from a separate indentation, nine different positions were measured near the center of each cell. Representative AFM images are shown in the right panel (scale bar 10 μm). **B** Superposition of the Young's modulus distribution (left panel) and box plot of the average value from the four different experiments (right panel). Boxes represent the 25–75 percentiles of the sample distribution, with black vertical lines representing the 1.5 × IQR (interquartile range). The black dots represent outliers. Black horizontal line represents the median. Significance was calculated using one-way ANOVA, followed by a Tukey post hoc test. For **A** and **B** three independent experiments were performed, each comparing naïve RBCs, and naïve RBCs treated with *Pf*-derived EVs in the presence or absence of bortezomib, with total of 15 full images acquired for each condition. Number of cells measured were for No EVs n = 86, *Pf*-derived EVs n = 103, *Pf*-derived EVs + bortezomib n = 95, p < 0.001. **C** Representative AFM images recorded in air showing the cytoskeletal structures of naïve RBCs, naïve RBCs treated with *Pf*-derived EVs in the presence or absence of bortezomib. Three independent experiments were performed, with a total of 50 AFM scans for each type of cell treatment. **D** Demonstration of the CNN analysis discrimination. Two examples are shown for each cytoskeleton phenotype: uRBCs on the left, RBCs exposed to *Pf*-derived EVs without and with bortezomib treatment in the center and right, respectively. Top row shows AFM images and bottom row the class activation maps whereby discriminatory regions are red/yellow and uncorrelated or anti-correlated regions blue/green. Discriminatory regions are marked and labeled to guide the eye. It is clear that the "hole" regions are associated with healthy untreated RBC and RBC treated with *Pf*-derived EV and bortezomib and the "backbone" regions with the RBC affected with *Pf*-derived EVs. **E** CNN prediction, showing analysis made on the control set of naïve RBCs, and naïve RBCs treated with *Pf*-derived EVs in the presence or absence of bortezomib (proteasome inhibitor). The *x*-axis represents the probability for an image to resemble the uRBC control. The *y* axis represents the number of images in the analysis for a class type (untreated RBC, *Pf*-derived EV-treated RBC, and RBC treated with *Pf*-derived EV and bortezomib) which conforms to a given probability. The no EV and *Pf*-derived EV data is from the testing set.

proteasome inhibitor-treated sample are indistinguishable ($p =$ 0.88). (Fig. 4A, B and Supplementary Fig. S12). These results imply that the 20S proteasome complexes shuttled within *Pf*-derived EVs are involved in modulating the deformability of the RBC.

Furthermore, we investigated whether the topological and morphological changes of the cytoskeleton network induced by *Pf*-derived EVs (as shown in Fig. 1D) are mediated by the activity of the delivered 20S proteasomes. This was achieved by treating *Pf*-derived EVs with and without proteasome inhibitor followed by incubation with host cells. RBCs were then imaged by high-resolution AFM. Here, too, following *Pf*-derived EV treatment, the fibrillar structure bunched together, leaving gaps in the cytoskeleton, i.e an increase in the mesh size of the cytoskeletal network (Fig. 4C middle panel). However, in the presence of the proteasome inhibitor, the *Pf*-derived EV-treated RBC sample showed a well-defined cytoskeleton (Fig. 4C, bottom panel), similar to that of control RBCs.

In order to objectively validate these morphological results regarding cytoskeleton disruption (as detected by 'holes' and filament 'mashing'), we analyzed the collected AFM images obtained from all treatments (by two independent experiments) using a convolutional neural network (CNN) deep learning model (see methods section). The natural variations arising from biological and local kinetic variability in extent of the damage complicated the identification of image features that could be definitively related to the changes. The CNN model removes analysis bias and lends confidence to the interpretation of our results in light of the significant variability between different cells. We trained the model on the control untreated RBCs and *Pf*-derived EVs treated AFM images. We then used the trained model as an independent approach to predict the type of treatment to which the RBCs were exposed on the testing set.

This analysis provided two important results. Firstly, it identified the surface features from the cytoskeleton images associated with healthy or treated cells. Secondly, it provided a quantitative measure of the cell's health. Figure 4D, E and Supplementary Fig. S13 presents this data. Figure 4D shows sample AFM images of healthy naïve RBCs, and *Pf*-derived EV-treated RBCs – the latter both untreated and treated with bortezomib – and the corresponding class activation maps. Comparing the maps for healthy naïve RBCs and *Pf*-derived EV-treated RBCs highlights the significant regions arising from the classification[45]. Within the limits of the reduced resolution inherent in these maps (see below), it is clear that the holes in the structure are discriminatory regions associated with the healthy cells and the backbone regions discriminatory for the treated ones. This identifies the transition from a semi-regular network of nodes that are connected by well-differentiated filaments surrounding holes, to a morphology where the holes have been reduced in size or in number, leaving a denser, amorphous background of membrane proteins, and without clearly differentiated filaments. Because the CNN process reduces the pixels used for each generation of feature maps, and the final weighted sum of the feature maps is upsampled to the original image size, we are unable to link the difference in data sets to high level tangible fine structural details such as the size and local arrangement of the individual fibrils. The quantitative specificity of two filters (activation of holes and backbone) is displayed in the distribution plots in Supplementary Fig. S13F, G.

Next, the full CNN analysis was used to assign the probability that a particular image belongs to the set of healthy or exposed cells. The closer the probability was to 1.0, the more closely the image resembled the control naïve RBCs (Fig. 4E and Supplementary Fig. S13B–E). The class activation maps and probabilities (Fig. 4D and Supplementary Fig. S13D, E) for

*Pf*-derived EVs treated with bortezomib, associate those phenotypes with the healthy cell. Two sets were tested: one gave a probability of 94% (Supplementary Fig. S13D) and the other of 86% (Supplementary Fig. S13E) of being similar to the control images (i.e., uRBCs), indicating that proteasome inhibition prevents cytoskeleton disruption. These quantitative results strongly support the involvement of *Pf*-EV-20S in membrane cytoskeleton remodeling of the host RBC.

**Phosphorylated cytoskeleton proteins are degradation substrates of the secreted EV-20S proteasome.** Having established that (I) *Pf*-derived EVs trigger unique phosphorylation events among proteins belonging to the cytoskeleton network (Fig. 2G and Supplementary Fig. S5), and that (II) the *Pf*-EV-20S proteasome is functional and is involved in inducing mechanical alterations in the RBC (Figs. 3 and 4). We subsequently directed our efforts to determine whether these cytoskeleton proteins are, in fact, direct substrates of the *Pf*-EV-20S proteasome. We, therefore, measured the stability of five cytoskeleton proteins, pre and post *Pf*-derived EV treatment and in the presence or absence of a proteasome inhibitor. Western blot analysis detected a significant decrease in the levels of β-adducin, ankyrin-1, dematin, and Epb4.1 following treatment with *Pf*-EV-20S proteasomes, whereas no changes were observed in the levels of spectrin α−chain (Fig. 5A, B and Supplementary Fig. S14A). In contrast, treating *Pf*-derived EVs with a proteasome inhibitor stabilizes the level of the four proteins, suggesting that they are target substrates of the *Pf*-EV-20S proteasome.

It is known that the 20S proteasome can degrade only proteins that contain partially unfolded regions[38,41], such as proteins harboring intrinsically disordered regions (IDRs), unstructured segments of more than 30 amino acids in length. Remarkably, our examination of the structural properties of the four substrates, using disorder prediction algorithms[46], generated by $D^2P^2$, revealed that at least 20% of each of the four substrates' residues is predicted to be IDRs, with β-adducin exhibiting 41%, ankyrin-1 20%, dematin 64%, and Epb4.1 39% disorder (Fig. 5C). Moreover, we noticed that the phosphosites that are unique to the *Pf*-derived EVs (Table 1) are all located within the unstructured regions, suggesting that they facilitate domain exposure that causes degradation (Fig. 5C). On the other hand, spectrin α−chain, which was not degraded by the *Pf*-EV-20S proteasome, displays only 2% disorder (Fig. 5A–C). Taken together, these results confirm that the four IDR proteins are susceptible to 20S proteasome-mediated degradation, suggesting that these critical components of the RBC cytoskeletal network are degradation targets of the *Pf*-EV-20S proteasome.

To determine whether 20S activity can alone degrade these proteins, irrespective of the other EV cargo components, we established an in vitro degradation system. In this system, the degradation levels of proteins are measured following treatment with purified 20S proteasome complexes[47,48]. After validating the activity of the purified 20S proteasome (Supplementary Fig. S15), 20S proteasome complexes were added to ghost RBCs, which are deprived of cytoplasmic proteins and, thus contain mainly the membrane fraction. We monitored the levels of dematin, β-adducin, ankyrin-1, and Epb4.1, using Western blot analysis before and after the addition of purified 20S proteasome complexes, in the presence or absence of the proteasome inhibitor. Notably, none of the proteins were degraded by the purified 20S proteasome complex, regardless of the proteasome inhibitor's presence (Fig. 5D and Supplementary Fig. S16). Thus, the results suggest that in order to transform these cytoskeletal proteins into mature substrates of the 20S proteasome, an additional component within the *Pf*-derived EVs is required.

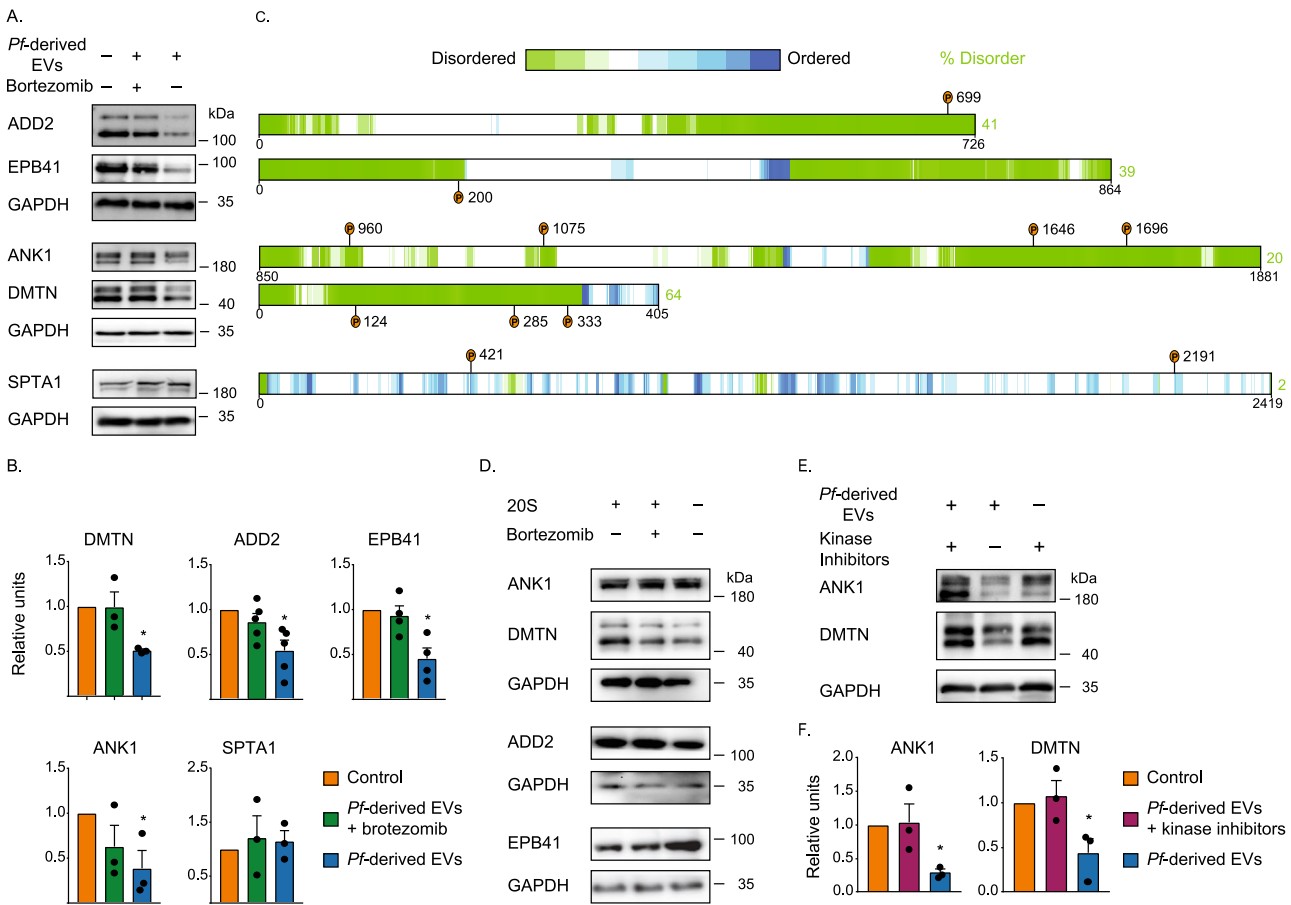

**Fig. 5 RBC cytoskeleton proteins are target substrates of the EV-20S proteasome. A** Degradation of cytoskeletal proteins following *Pf*-derived EV treatment. Western blot analysis of naïve RBCs treated with *Pf*-derived EVs in the presence or absence of the proteasome inhibitor bortezomib. Commercial antibodies were used against five cytoskeletal proteins: β-adducin (ADD2), erythrocyte membrane protein band 4.1 (EPB4.1), ankyrin-1 (ANK1), dematin (DMTN), and spectrin α−chain (SPTA1). For each gel, GAPDH was used as a loading control. **B** Quantification and averaging of three independent experiments (as in **A**). Each protein's value was divided by the control value for that batch, thus creating a pair of ratios for each batch. The ratios of *Pf*-derived EVs were tested against the ratios of the *Pf*-derived EVs + bortezomib using a paired two-way *t*-test, error bars represent SD (DMTN *p = 0.0102, ADD2 *p = 0.01339, EPB41 *p = 0.02769, ANK *p = 0.01339). **C** Graphical illustration of the disorder prediction of human β-adducin, Ebp4.1, ankyrin-1, dematin, and spectrin α−chain, generated by D²P² (website: http://d2p2.pro), a database of protein disorder predictions. The level of disorder prediction is shown as a color intensity in which green segments represent disordered regions. The position of the unique phosphosites following treatment with *Pf*-derived EVs is indicated by orange circles. **D** Degradation assay of ghost naïve RBCs with purified 20S proteasome complexes. Bortezomib was used as a control for 20S proteasome inhibition. Panels display immunoblots using the relevant antibodies. Results represent three independent experiments (Supplementary Fig. S16). **E** Degradation of cytoskeletal proteins following *Pf*-derived EV treatment in the presence of kinase inhibitors mixture. Western blot analysis for naïve RBCs that were treated with *Pf*-derived EVs in the presence or absence of kinase inhibitor mixture (staurosporine and dihydrochloride). Antibodies were used against ankyrin-1 and dematin. GAPDH was used as a loading control (Supplementary Fig. S17).
**F** Quantification and averaging of three independent experiments (as in **E**), using a paired two-way *t*-test, error bars represent SD (ANK *p = 0.0421, DMTN *p = 0.0402). Source data for **A** and **D** are provided in the source data file. Source data for **B** and **F** are provided in Supplementary Fig. S14.

Since the cytoskeletal substrates acquire unique phosphorylation sites upon treatment with *Pf*-derived EVs (Figs. 2B, E and 5C and Table 1), we reasoned that this phosphorylation step might be necessary for shifting the proteins toward 20S proteolysis. This hypothesis is supported by the fact that the identified unique phosphorylation sites are all located within the unstructured regions, suggesting that they facilitate conformational transitions that cause degradation (Fig. 5B, C). To test this assumption, we treated naïve RBCs with *Pf*-derived EVs in the presence and absence of a mix of two known kinase inhibitors (staurosporine and dihydrochloride) and examined the level of two representative proteins, dematin and ankyrin-1. We found that both proteins were clearly stabilized upon cell treatment with *Pf*-derived EVs in the presence of these kinase inhibitors (Fig. 5E, F and Supplementary Fig. S14B and S17), suggesting that the phosphorylation serves as a precondition step prior to

proteasome degradation. Collectively, our results indicate that RBC cytoskeleton proteins are direct degradation substrates of the delivered 20S proteasome and that this degradation is facilitated by delivery of *Pf*-derived EV kinases.

## Discussion

Here we discovered a mechanism by which malaria parasites use EVs to facilitate their growth in their most essential host cell, the RBC (Fig. 6). In particular, we showed that assembled and active 20S proteasome complexes are delivered together with kinases to naïve RBCs through EVs that are released from *Pf*-infected RBCs. This leads to a 20S-proteasome-dependent process of cytoskeleton network remodeling, which eventually enhances parasitemia. We found that following *Pf*-derived EV introduction, two sequential steps take place. First, RBC proteins are subjected to

specific phosphorylation, including many cytoskeletal proteins. Then, the delivered 20S proteasome mediates the degradation of phosphorylated cytoskeleton proteins by a ubiquitin-independent process. This mechanism, reduces the stiffness of the membrane, and therefore primes naïve RBCs for parasite invasion, eventually increasing the parasite's growth capacity. Put together, our results combine to reveal a previously unidentified pathway of Pf survival in the human host RBC.

The parasite invasion involves overcoming the barrier of bending the host membrane as it wraps the parasite[49], and production of biophysical forces between the two cells: the parasite and the host RBC[50]. Here, we show that the parasite while growing inside the RBC, uses secreted 20S complexes to modify the biophysical properties of 'future' host cells before invasion, to increase membrane deformability for its benefit. Since increased deformability of the RBC is required for merozoite invasion[6], we suggest that the secreted EV-20S complexes assist the parasite in a similar manner by increasing 'in advance' the host deformability of surrounding cells. Our observations indicate that the EV-20S proteasomes degrade the protein complexes that are involved in binding the membrane to the spectrin filaments, which form the backbone of the RBC cytoskeletal network, which in normal RBCs is adsorbed to the membrane and spans the whole surface of the cell. Upon degradation of these complexes, the spectrin network connectivity is broken, with spectrin filaments only partially bound to the membrane. These biochemical observations therefore naturally explain the reduced stiffness[51] as indicated by the AFM measurements, and the observed damage to the network organization (Figs. 1 and 4). These observed changes in naïve RBCs are in accordance with previous reports that, within blood samples from malaria patients, the uRBCs surrounding iRBCs exhibit alterations and accelerated cell destruction[52,53]. The new observations also offer a natural explanation for the role of the ATP content of the RBC during the parasite invasion, with lower invasion in low-ATP-containing RBCs[51,54].

We stress that the changes studied here are preparatory steps which condition the prospective host cell at the "pre-invasion" stage. The literature contains various studies on stiffening and softening of damaged RBCs in general, and those infected with the Pf parasite in particular[6,7]. In all of these cases, damage both by chemical agents and by biological processes enhanced the rate of infection following softening of the RBC, whereas infection was inhibited when the cell was stiffened in qualitative agreement with our findings. Our work provides a detailed insight into the specific nature of the biochemical process occurring and the mechanical implications, related to the different stages of the entire infection process.

The parasite's specific preference for the 20S proteasome is interesting, given that the degradation of proteins is predominantly mediated by the ubiquitin-dependent 26S proteasomal pathway[55], which comprises the 20S catalytic core particle and the 19S regulatory complex. The existence of the alternative, simpler degradation route mediated solely by the 20S proteasome is a recent discovery, (reviewed in refs. [38,41]). Unlike its 26S counterpart, wherein substrate selectivity is achieved by ubiquitin tagging, the 20S proteasome cleaves proteins containing partially unfolded regions that can enter into its catalytic chamber. Although this field of research is still in its infancy, accumulating evidence indicates that the 20S proteasome degradation route is tightly regulated[47], influencing various cellular processes, such as neuronal stimulation[56,57], antigenic peptide production[58,59], and post-translational processing[60–64]. When considering the requirements of 26S proteasome-mediated degradation, i.e., the dependency on ATP hydrolysis and ubiquitinylating enzymes (E1, E2, and E3), the preference of the parasite to harness the 20S-mediated degradation route is reasonable. Another critical factor

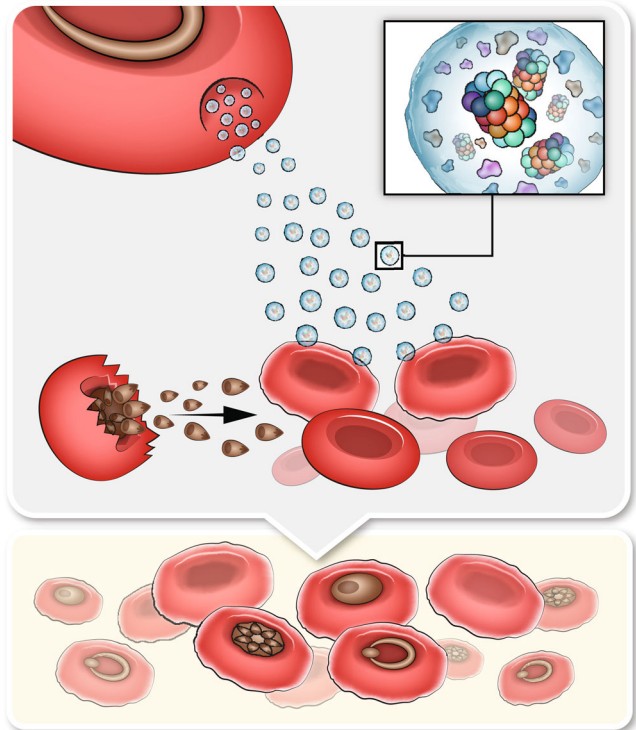

**Fig. 6 Pf-derived EVs prime naïve host RBCs to enhance parasite growth.** Diagram demonstrating the impact of Pf-derived EVs on naïve RBCs. Our results indicate that EVs enriched with active proteasome complexes and protein kinases invade naïve host RBCs. This process leads to the phosphorylation and subsequent degradation of RBC cytoskeleton proteins. Consequently, the cytoskeleton network of the recipient naïve RBC is disrupted and the cell membrane is remodeled, improving the invasive capacity of the malaria parasite.

is the restricted size of the malaria-derived EVs, 50–200 nm in diameter, which limits the encapsulation of 26S proteasomes (~45 × 20 nm), to a greater degree than 20S particles (~15 × 12 nm). Thus, degradation via the 20S proteasome offers a relatively compact system for eliminating proteins, one that reduces energy costs and is not dependent on an enzymatic cascade.

Considering the unique anuclear configuration of mature RBCs, an attempt at altering the conformation, function or integrity of RBC proteins can be achieved via protein post-translation modification (PTM) mechanisms. Because PTMs influence protein activity, folding, interactions, localization, stability and turnover, they are key in defining the protein structure-function relationship in every living organism, including in malaria parasites[65]. Among all PTM mechanisms, phosphorylation is perhaps the most well-studied to date in Plasmodium with numerous parasite and host cell protein kinases identified as essential for the development of the parasite throughout its entire life cycle[66]. Interestingly, many PTMs, including phosphorylation, are increasingly being shown to modify proteins harboring IDRs[67], protein regions lacking stable tertiary and/or secondary structure, but characterized by highly dynamic structures extremely sensitive to changes in the environment[66]. Indeed, we found that the EVs secreted from Pf-iRBCs carry a large group of parasitic kinases (14 Pf kinases, Fig. 2B), of which many could potentially be involved in the phosphorylation of host membrane and cytoskeletal proteins. The observation that in addition to the delivery of 20S proteasomes, parasitic kinases are also delivered by the Pf-derived EVs, is reasonable given that under basal conditions 20S proteasomes are found both within uRBC and

EVs derived from uRBCs while maintaining the integrity of the host's cytoskeleton proteins. Thus, the parasite uses the kinases to specifically phosphorylate RBC cytoskeleton proteins. This, in turn, primes the target proteins towards degradation and, consequently, to cytoskeleton destabilization. Interestingly, and in line with our data, it was previously shown that *Pf* parasites secrete protein kinases outside of the iRBCs through an unknown mechanism, and it was suggested that EVs could play a role in this mechanism[68,69].

Protein kinases are considered to be excellent drug targets for a variety of human diseases. To the best of our knowledge, only two *Plasmodium* protein kinases, *Pf*CK1[68] and *Pf*TL2[69], have been shown to date to be secreted by the iRBC to the extracellular milieu. Although *Pf*CK1 and *Pf*TKL2 were not identified in the proteomics analysis of parasite derived EVs, other protein kinases have been detected and remain likely candidates responsible for the observed increased phosphorylation of RBC cytoskeletal proteins. Inhibition of the kinase activity detected in *Pf*-derived EVs will be an exciting avenue for testing the potential of specific kinase inhibitors in preventing parasite expansion.

Another avenue for therapeutic intervention is the 20S proteasome. The malaria proteasome degradation pathway is considered a high-value druggable target due to its essential role in protein turnover and the parasite's need to rapidly divide inside host cells[70]. This was recently confirmed by a study demonstrating the high ratio of essential or dispensable genes in the parasite proteasome system[71]. However, to date, most compounds tested against the parasite also inhibit the mammalian proteasome, resulting in toxicity that precludes their use as therapeutic agents[72]. Nevertheless, recent structural studies have concluded that the *Pf*-20S proteasome is sufficiently unique from the human proteasome for selective targeting[73,74]. Taken together with our findings, we anticipate that efficient inhibition of the EV-proteasome-dependent RBC priming process could serve as an avenue for therapeutic treatment. To realize such an approach, however, a first critical step is to determine whether the *Pf*-derived EV-delivered 20S proteasome is composed solely of human subunits, parasite components or a hybrid of the two.

We demonstrated that functional cellular machinery, the 20S proteasome, is transported by EVs in order to reshape the host membrane. Along these lines, recent proteomic data indicated the presence of subunits of the 20S proteasome in EVs derived from many pathogens, including pathogenic protozoans such as *Acanthamoeba castellanii*[75] and *Toxoplasma gondii*[76], helminth parasites such as *Echinococcus granulosus*[77], as well as *Leishmania*[78], *Trichuris muris*[79], *Trichomonas vaginalis*[80], and even fungus[81]. These results may therefore hint to a more general mechanism, in which pathogen-derived EVs use the degradation machinery to reshape the host proteome[49].

From a broader perspective, active 20S proteasomes, and not 26S proteasomes, were recently discovered within EVs secreted from tumor-associated macrophages[82], mesenchymal stem cells[83], and endothelial cells[84]. Hence, it is possible that EV-mediated delivery of functional 20S proteasome complexes for cellular remodeling is a more common phenomenon then currently believed.

## Methods

**Parasite line and culture**. The NF54 parasite line is a common human infecting *Pf* line and was obtained from the Malaria Research Reference Reagent Resource Center (MR4). Parasites were grown in pooled donor RBCs provided by the Israeli blood bank (Magen David Adom blood donations in Israel) at 4% hematocrit, and incubated at 37 °C in a gas mixture of 1% $O_2$, 5% $CO_2$ in $N_2$. In order to culture the parasite, we used human RBCs that were pooled from approximately total of 25 different healthy donors. Parasites were maintained in RPMI medium pH 7.4, 25 mg/ml HEPES, 50 μg/ml hypoxanthine, 2 mg/ml sodium bicarbonate, 20 μg/ml gentamycin, and 0.5% (w/v) AlbumaxII (Invitrogen), as previously described[85].

*P. falciparum* cultures were tested for mycoplasma once a month using MycoAlert™ PLUS (LONZA, Cat# LT07-703).

**Magnetic isolation of *Pf* -iRBCs**. iRBCs were purified using a magnetized MACS column as described in[86] with minor modifications. Briefly, the iRBC culture was loaded on a pre-washed MACS© CS column (Milteny Biotech, Cologne, Germany). Infected cells were eluted at a rate of one drop per one second, washed with 10 column volumes of pre-warmed RPMI and eluted to a fresh 50 mL tube.

**EV isolation**. EVs were isolated from either the NF54 strain in a relatively high *P. falciparum*-iRBC parasitemia level (~8%) at 4% hematocrit and highly synchronized parasite culture, 24 h post invasion into the RBC (Trophozoite stage), or from uRBC culture. We extracted EVs using a Beckman OPTIMA90X ultracentrifuge with a TI70 rotor, as previously described[34]. Briefly, 200–400 ml of a parasite growth medium was collected and then cellular debris were removed by centrifugation at 413 × *g* for 5 min, 1650 × *g* for 10 min (Eppendorf Centrifuge 5804), followed by centrifugation at 15,180 *g* for 1 h in a SORVALL RC5C PLUS, SLA-1500 rotor. The supernatant was filtered through a 0.45-μm filter, concentrated down using a VivaCell 100,000 MWCO PES (Sartorious Stedium) and ultracentrifuged at 150,000 × *g* for 18 h to pellet EVs. The pellet was resuspended in PBS−/− or in a cell culture medium. In all, 100 μL suspension of EVs yields a typical concentration of $1 \times 10^{12}$ particles per ml. Purified EVs were added into medium of growing *Pf*-iRBCs culture.

**Nanoparticle tracking analysis**. EV size and concentration distribution was performed using nanoparticle tracking analysis (Malvern Instruments Ltd., NanoSight NS300) performed at 20 °C[19]. Sample size distributions were obtained in a liquid suspension (1:1000 dilution in PBS−/−) by the analysis of Brownian motion via light scattering. The camera level was set to 13 and the gain to 1, blue (405 nm) or green (488 nm) lasers without filter, following the manufacturer's instruction. The data was analyzed by using NTA 2·1 software (NanoSight).

**OptiPrep gradient fractionation**. EVs were fractionated by centrifugation (250,000 × *g*, 18 h, 4 °C) in a SW41 rotor (Beckman Coulter, Fullerton, CA, USA) through a continuous 5–40% OptiPrep (Sigma-Aldrich, cat# D1556) gradient. Fractions (1 ml) were collected from the top of the gradient for further analysis.

**Parasite growth assay and immunofluorescence labeling**. To monitor parasitemia levels, *Pf*-iRBCs were collected and incubated with a mixture of nucleic acid stain dyes: The DNA dye Hoechst 33342 (Invitrogen cat# H1399) was titrated to 5 μM and the RNA dye Thiazole Orange (Sigma-Aldrich, cat# 390062) was diluted 1:100,000 from a 1 mg/ml stock. *Pf*-iRBCs were incubated for 30 min at 37 °C and were analyzed in a Bio-Rad ZE5™ flow cytometer. Parasitemia (rings) was counted 3 or 5 days after infection by microscopy of Giemsa stained thin smears, using an upright NIKON objective 100x Plan Apo and NIS Element software version 4.4.

For parasite growth assay in the presence of bortezomib-treated EVs: Medium harvested from uRBC culture containing EVs was treated with 0.05 μM bortezomib, at 37 °C for 2 h. Following this step, the treated EVs were washed with PBS via ultracentrifugation spin ON. uRBC-derived EVs with or without bortezomib were introduced to naïve RBCs for 18 h. Following RBC pretreatment, magnet-purified late stage-parasites (0.5%) were introduced and parasitemia was monitored after 12 h (ring stage) using flow cytometry (Biorad-ZE5™). The data was analyzed by using Everest v. 2.3.03.0 software (BioRad). The FACS gating strategy is demonstrated in Fig. S18.

**Preparation of large unilamellar vesicles**. LUVs were prepared with a lipid composition of DOPC:DOPE:DOPS:SM:chol of 20:5:15:25:35 (molar ratio), resembling the RBC plasma membrane composition as previously described[33]. Lipid solution in chloroform of the different phospholipid species were mixed to the desired molar ratios in a glass vial, and organic solvent was evaporated by 12 h of vacuum pumping. For labeled LUVs, the lipids were stained with 2% mol of DiI and DiD in chloroform before evaporation. The lipid film was then hydrated with PBS (Ca$^{2+}$-/Mg$^{2+}$-at 40 °C to reach the desired concentration and gently vortexed. The resulting MLV suspension was then sonicated for 10 min to disperse larger aggregates and the liposomal suspension was extruded 21 times through polycarbonate filters (100 nm pore size, Avanti Polar Lipids) using a mini-extruder (Avanti Polar Lipids). Size and concentration were verified using NTA and the liposomal suspension was used within 2 weeks from the extrusion. DiI (1,1′-dioctadecyl-3,3,3′,3′-tetramethylindocarbocyanine perchlorate) and DiD (1,1′-Dioctadecyl-3,3,3′,3′-Tetramethylindodicarbocyanine 4-Chlorobenzenesulfonate Salt) membrane dyes were purchased from Thermo Fisher as a powder and dissolved in chloroform at 1 mM final concentration. All chemicals had high purity and were used without further purification.

**Membrane mixing assay**. All experiments were performed using a Cytation 5 Imaging Reader plate reader (BioTek) with a 96-well plate. DiI-DiD labeled liposomes were diluted in 200 μl PBS (Ca$^{2+}$-/Mg$^{2+}$-) per well to reach a final concentration of 10 μM, and fluorescence intensity of the donor (DiI) was recorded

every 60 s for 30 min, with excitation wavelength of 530 nm and emission wave-lengths of 570 nm. Subsequently, unlabeled $Pf$ – derived EVs or unlabeled LUVs were added in each well to reach a labeled: unlabeled ratio of 1:9 particles and DiI fluorescence intensity was recorded for 1 h every 60 s. Finally, Triton X-100 was added in each well to reach 0.1% final concentration and fluorescence intensity was recorded for 15 min every 60 s. The emission fluorescence for each time point was measured as $I_n$. The emission fluorescence of the untreated liposomes was measured as $I_0$, and that of the liposomes solubilized with 0.1% TRITON X-100 was defined as $I_{100}$. The percentage of membrane mixing at each time point is defined as: donor relative intensity (% of TRITON X-100) = $(I_n − I_0) \times 100/(I_{100} − I_0)$. All measurements were performed at 37 °C. The data was analyzed by using Gen5™ v. 3.04 software (BioTek).

**Size distribution NTA analysis for EV fusion events**. LUVs and $Pf$-derived EVs were mixed in a 1:1 ratio (particles: particles) to a final concentration of $1–10 \times 10^8$ particles/mL in 1 mL of filtered PBS ($Ca^{2+}$-/$Mg^{2+}$-) and size distribution was immediately measured using a NanoSight NS300. Briefly, approximately 1 ml solution was loaded into the sample chamber of an LM10 unit (NanoSight) and five videos of 60 s were recorded. Data analysis was performed with NTA 2·1 software (NanoSight). This size distribution was considered as time 0 min of the interaction between LUVs and EVs. Subsequently, EVs from the same biological sample were mixed with LUVs at the same particle ratio and incubated at 37 °C for 30 min, after which size distribution was measured using the same recording parameters and detection threshold of the sample at time 0 min. The resulting size distribution curves were then analyzed by summing for each individual technical repeat of the acquisition the number of particles above the D90 parameter value (the diameter where ninety percent of the distribution has a smaller particle size and ten percent has a larger particle size) obtained at time 0 min, and the fraction of particles above D90 was calculated as: number of particles above D90/ total number of particles. The statistical significance of the fraction above D90 at time 0 between the two different time points was calculated with a two-sided t-test with a p-value of 0.05.

**Protein extraction from naïve red blood cells**. Naïve human RBCs were washed with iso-osmotic buffer (103 mM $Na_2HPO_4$, 155 mM $NaH_2PO_4$, Sigma-Aldrich), followed by lysis with RIPA buffer (150 mM NaCl, 1% Triton X-100, 0.1% SDS, 50 mM Tris, 0.5% Sodium deoxycholate, pH 8.0) with or without a protease inhibitor mix (1.5 mM Aprotinin (A1153), 1.5 mM Pepstatin A (P5318), 2 mM Leupeptin (L2884), 0.5 mM PMSF (P7626) (Sigma-Aldrich)) and Halt™ Phosphatase Inhibitor Cocktail (78420, Thermo Fisher scientific). Cells were incubated with the RIPA buffer for 15 min on ice, followed by 10 min centrifugation on the maximum speed at 4 °C. The supernatant was collected and hemoglobin was depleted using TALON metal affinity beads (635502 Takara-Clontech). For ghost purification, the pellet was re-washed with 1 mL iso-osmotic buffer until the membrane pellet became clear.

**Western blot assay**. Proteins were separated on 10–15% SDS-PAGE and transferred to nitrocellulose membranes. The primary antibodies used for detection were obtained commercially from Abcam®. anti-dematin (ab226357, dilution 1:1000), anti-adducin 2 (beta-adducin) (ab251821, dilution 1:500), anti-Epb4.1 (ab185704, dilution 1:1000), anti-ankyrin-1 (S288A-10, dilution 1:1000) (ab212053), anti-HSP90 (AC88) (ab13492, dilution 1:10,000), anti-GAPDH (6C5) (ab8245, dilution 1:5000), anti-alpha 1 spectrin (17C7) (ab11751, dilution 1:1000), anti-SR1 (ab71983, dilution 1:1000), anti-phospho Ser/Thr, anti-Phe (ab17464, dilution 1:1000), anti-PSMD1 (ab2941, dilution 1:5000), anti-PSMA1 (ab140499, dilution 1:5000) and anti-ubiquitin (PW0930, Enzo, dilution 1:1000). Secondary antibodies used are Goat anti Rabbit IgG-HPR (111-035-003, Jackson, dilution 1:10000), and Goat anti mouse IgG-HRP (115-035-003, Jackson, dilution 1:10,000). Western blot analyses were repeated at least three times. Image collection was performed using the ThermoScientific MyECL Imager V. 2.2.0.1250 and Amersham Imager 680 (GE Healthcare Life Sciences). Quantifications of the blots were done using ImageJ 1.51k, R v.4 software.

Analysis of total ubiquitination in RBCs and EVs was performed as follows: naïve RBCs ($~1 \times 10^8$ cells) were incubated with $Pf$-derived EVs and control EVs (40 µl of EVs at a concentration of $2 \times 10^{11}$ particles/ml) and analyzed for total ubiquitination by Western blot analysis using an anti-ubiquitin antibody. Total ubiquitination in EVs was similarly monitored in extracts from 50 µl of EVs at a concentration of $2 \times 10^{11}$ particles/ml).

**Kinase activity for *P. falciparum*-derived EVs**. Equal volumes of EVs were sonicated using BRANSONIC 5 for 10 s with ATP in kinase reaction buffer (20 mM Tris-HCl pH 7.5, 10 mM $MgCl_2$) prior to 30 min incubation. Sample buffer was immediately added and the mixture heated to 60 °C. Samples were subjected to 10% SDS-PAGE gel separation.

**Prediction of protein unstructured regions**. To predict unstructured regions in cytoskeleton proteins, we analyzed the protein sequence using the Database of Disordered Proteins Predictions (http://d2p2.pro/). The consensus sequence (>75%) among the nine different prediction algorithms used by the software is displayed.

**20S proteasome in-gel activity assay**. The catalytic activity of the 20S and 26S proteasomes was assayed as we have previously done[87]. Briefly, 20 µg/µl of RBC lysates or extracts from $1 \times 10^{11}$ EVs were separated on 4% native-PAGE using native running buffer (0.09 M Tris-base, 0.09 M boric acid, 1 mM EDTA, 2.5 mM $MgCl_2$, 0.5 mM ATP and 1 mM DTT). Gels were soaked in 10 ml native buffer supplemented with 100 µM of the proteasome fluorogenic peptide substrate Suc-LLVY-AMC (Bachem I-1395.0100) and 0.02% SDS. Gels were incubated in the dark at 30 °C for 15–30 min and then imaged under ultraviolet light. Proteasome subunits were further verified by western blot using an anti-PSMA1 antibody and an anti-PSMD1 antibody.

**Proteasome activity assay in solution (peptidase assay)**. To monitor 20S proteasome activity, we measured the hydrolysis of 100 µM of the proteasome fluorogenic peptide substrate, Suc-LLVY-AMC, as done previously[87]. Briefly, 20 µl vesicles at a concentration of $1.5–2 \times 10^{12}$ particles per ml were sonicated in a bath sonicator for 10 s and then incubated with the fluorescent peptide for 1 h in the dark at 37 °C. After incubation, the fluorescence of the hydrolyzed AMC groups was measured with a microplate reader (Infinite 200, Tecan Group, Tecan icontrol v. 3.9.1.0), using an excitation filter of 380 nm and an emission filter of 460 nm.

Time course experiments were performed in a similar manner, using three fluorescent substrate peptides for the three catalytic activities of the 20S proteasome; Suc-LLVY-AMC for the chymotrypsin-like activity, Z-Leu-Leu-Glu-AMC (BML-ZW9345, Enzo) for the caspase-like activity and Boc-Leu-Arg-Arg-AMC (BML-BW8515, Enzo) for the trypsin-like activity. Measurements were taken at 1 min intervals over a period of 90 min. During the measurement, the plate was kept at 37 °C.

**Cytoskeleton protein degradation assay**. $Pf$-derived EVs were produced as described above. Following the VivaCell filtration, media containing EVs was incubated with 0.05 µM bortezomib (Merck 5.04314.0001) at 37 °C for 2 h. Naïve RBCs were incubated with $Pf$-derived EVs with or without pretreatment with bortezomib, for 3 h. The treated cells were then lysed and separated by 10% SDS-PAGE. Protein amounts were evaluated using a Western blot assay.

In vitro degradation assay of cytoskeleton proteins were done by incubating $~7$ µM of purified 20S proteasomes with RBC ghost cells (8 mg/ml total protein) for 3 h at 37 °C in the presence or absence of the 20S proteasome inhibitor bortezomib (0.05 µM). Samples were then separated using 12% SDS-PAGE and protein amounts were evaluated using Western blots.

**Cytoskeleton protein degradation assay in the presence of kinase inhibitors**. In all, 40 µl of EVs (in a concentration of $2 \times 10^{11}$ particles/ml) were incubated with or without kinase inhibitor mixture (14 nM (R)-DRF053 dihydrochloride, D6946-5MG, Sigma Aldrich, 1 µM Staurosphorine, (kindly provided by G-INCPM, Weizmann Institute) (v/v) for 1.5 h at 37 °C. Treated EVs were incubated for 3 h with naïve RBCs, followed by protein extraction (as described above) and separation on 10% SDS-PAGE. As controls, cells were incubated with kinase inhibitor mixture alone.

**Proteomics analysis of *Pf*-derived EVs**. TPCK-treated soybean trypsin and trypsin inhibitor were purchased from Worthington Biochemical Corporation. Parasites were grown to the trophozoite stage and then gelatin-floated for 45 min to enrich for iRBCs. $1 \times 10^8$ iRBCs and control uRBCs were digested with 1 mg/ml trypsin or PBS vehicle control at 37 °C for 1 h. Next, 4 mg/ml trypsin inhibitor was added and the samples were incubated at room temperature for 15 min. Samples were then lysed with 1% Triton-X 100 (Sigma) in PBS with Protease Inhibitor Cocktail (Roche) on ice for 20 min, centrifuged, and the supernatant was discarded. The pellet was washed with 1% Triton-X 100 and solubilized with 2% SDS in PBS with protease inhibitors. EVs were digested with 2 µg/ml trypsin for 15 min at 37 °C, in the presence or absence of 0.1% Triton-X 100, prior to the addition of 1 mg/ml trypsin inhibitor.

Tryptic peptides were separated by nano-flow reversed-phase liquid chromatography on a nanoACQUITY UPLC system (Waters, USA). Samples were loaded onto a 20-mm pre-column with 180 µm I.D. and 5 µm $C_{18}$ silica bead in 5% buffer B at a flow rate of 10 µl/min and then resolved on a 100-mm analytical column with 75 µm I.D. and 1.7 µm $C_{18}$ silica beads using a 90-min gradient set at a flow rate of 0.4 µl/min from 95% solvent A (0.1% formic acid in Milli-Q water) to 60% solvent B (0.1% formic acid, 80% acetonitrile in Milli-Q water). MS data were acquired on a Q-Exactive mass spectrometer fitted with a Proxeon nano-ESI source (Thermo Fisher Scientific). High mass-accuracy MS data was obtained in a data-dependent acquisition mode with the Orbitrap resolution set to 75,000 and the top-10 multiply charged species selected for fragmentation by HCD (single charged species were ignored). The ion threshold was set to 15,000 counts for MS/MS. The CE voltage was set to 27 V.

**Data analysis and functional annotation of proteomic data**. Mass spectra peak lists (MGF files) were extracted from the LC-MS/MS raw data using the software package ProteomeDiscoverer v. 2.1 (Thermo Fisher Scientific). For peptide identification and protein inference, the MGF peak lists were searched against a composite non-redundant protein sequence database comprising common

contaminants and *Homo sapiens* and *Plasmodium falciparum* sequences downloaded from UniProtKB. Mascot v. 2.4 (www.matrixscience.com) was used as the primary database search engine. In-house software tools (MSPro and Digger, v. 1.0; http://repository.unimelb.edu.au/10187/18167) were used to collate and merge the peptide and protein information across biological replicates. The database search parameters consisted of carbamidomethyl of cysteine as a fixed modification (+57 Da) and NH$_2$-terminal protein acetylation (+42 Da), N-terminal Q->pyroglutamic acid (−17 Da) and oxidation of methionine (+16 Da) as variable modifications. A precursor mass tolerance of ±10 ppm, #13 C defined as 1 and a fragment ion mass tolerance of ±20 ppm was used. The specified enzyme was trypsin/P, allowing for up to two missed tryptic cleavage sites. The program MSPro was used as a wrapper for parsing Mascot search result files and instantiating the post-processing software tool Percolator v.2.10[88]. The search results from the two independent biological replicates were analyzed individually and in combination in order to generate an inferred protein list for both *Plasmodium falciparum* and *Homo sapiens* (host) based on significant scoring (posterior error probabilities ≤1%) peptide identifications. Significant peptide matches were further categorized as unique or degenerate on the basis of peptide sequence and assigned to protein groups based on the principle of parsimony (Occam's razor). The inferred proteins for *Homo sapiens* and *Plasmodium falciparum* were submitted to The Gene Ontology Resource (http://geneontology.org)[89,90] and DAVID Bioinformatics Database (http://david.ncifcrf.gov)[91,92] for functional annotation and enrichment analysis. The LC-MS/MS raw data were also analyzed using MaxQuant v1.6.14 using default parameters for Thermo Q-Exactive data against the identical FASTA sequence database. The mass spectrometry proteomics data have been deposited to the ProteomeXchange Consortium via the PRIDE[93] partner repository with the dataset identifier PXD023353 and 10.6019/PXD023353.

**Phosphoproteomic analysis of EV-treated RBCs.** Naïve RBCs were treated with EVs derived from *Pf*-iRBCs and from uRBCs for two time points 5 and 15 min. Ghost samples were harvested and the pellet was subjected to mass spectrometry analysis. The samples were dissolved in 10 mM DTT, 100 mM Tris, 5% SDS, boiled at 95 °C for 5 min, and subjected to two cycles of sonication. The samples were precipitated in 80% acetone and washed three times with 80% acetone. Protein pellets were dissolved in 9 M Urea and 400 mM ammonium bicarbonate and then reduced with 10 mM DTT (60 °C for 30 min), modified with 5 mM iodoacetamide in 100 mM ammonium bicarbonate (room temperature for 30 min in the dark) and digested in 2 M Urea, 25 mM Tris with modified trypsin (Promega) at a 1:50 enzyme-to-substrate ratio overnight at 37 °C. An additional, second trypsinization was done for 4 h.

The tryptic peptides were desalted using C$_{18}$ tips (Oasis), dried and resuspended in 0.1% formic acid. Twenty percent of the resulting tryptic peptides were analyzed by LC-MS-MS. The remaining 80% of peptides (in 40% acetonitrile, 6% TFA) were enriched for phosphopeptides on titanium dioxide (TiO$_2$) beads. Titanium beads were pre-washed (80% acetonitrile, 6% TFA), mixed with the peptides for 30 min at room temperature, and then washed with 30% acetonitrile with 3% TFA and then with 80% acetonitrile with 0.1% TFA. Bound peptides were eluted with 20% acetonitrile with 325 mM ammonium hydroxide, followed by 80% acetonitrile with 325 mM ammonium hydroxide. The resultant peptides were desalted using C$_{18}$ tips and analyzed by LC-MS-MS.

In all, 2 μg of the resultant peptides were analyzed by LC-MS/MS using a Q-Exactive plus mass spectrometer (Thermo) fitted with a capillary HPLC (easy nLC 1000, Thermo-Fisher). The peptides were loaded onto a C$_{18}$ trap column (0.3 × 5 mm, LC-Packings), connected online to a homemade capillary column (20 cm, 75 μM ID), packed with Reprosil C$_{18}$-Aqua (Dr. Maisch GmbH, Germany) in solvent A (0.1% formic acid in water). The peptide mixture was resolved with a 5–28% linear gradient of solvent B (95% acetonitrile, 0.1% formic acid) for 120 min, followed by a 5-min linear gradient from 28 to 95% of solvent B and then 25 min at 95% acetonitrile with 0.1% formic acid in water at a flow rate of 0.15 μl/min. Mass spectrometry was performed in a positive ion mode (350–1800 *m/z* mass range, 70,000 resolution) using repetitively full MS scan followed by collision induced dissociation (HCD, 35 normalized collision energy) of the 10 most dominant ions (>1 charge) selected from the first full MS scan. MS data were acquired using the Dionex Chromatography MS Link (ThermoScientific) and Xcalibur v. 4.2 (ThermoScientific) software. The mass spectrometry phosphoproteomics data have been deposited to the ProteomeXchange Consortium via the PRIDE partner repository with the dataset identifier PXD018154.

**Phosphoproteomic data analysis.** The mass spectrometry data was analyzed using the MaxQuant software 1.5.2.8. (www.maxquant.org) for peak picking identification and quantitation using the Andromeda search engine, searching against the Human and *Pf* Uniprot database with a mass tolerance of 20 ppm for the precursor masses and 20 ppm for the fragment ions. Methionine oxidation, phosphorylation (STY) and protein N-terminus acetylation were accepted as variable modifications and carbamidomethyl on cysteine was accepted as a static modification. Minimal peptide length was set to six amino acids and maximum of two missed cleavages was allowed. Peptide- and protein-level false discovery rates (FDRs) were filtered to 1% using the target-decoy strategy. The protein table was filtered to eliminate identifications from the reverse database and common

contaminants. The data was quantified by label-free analysis using the MaxQuant software, based on extracted ion currents (XICs) of peptides, enabling quantitation from each LC/MS run for each peptide identified in any of the experiments.

**Immunoprecipitation of dematin.** Packed RBCs were incubated with equal amounts of *Pf*-derived EVs or uRBC-derived EVs for 30 min at 37 °C. Following incubation, the cells were lysed in RIPA buffer, hemoglobin was depleted, and 15% of the total lysate was taken as control. Immunoprecipitation using a dematin antibody was performed following the manufacturer's protocol of Bio-Rad SureBeads™ Protein G magnetic beads protocol. Samples were then separated by SDS-PAGE.

**Sample preparation for cytoskeleton AFM imaging.** Freshly cleaved mica surfaces were incubated with 0.1% polyLysine solution for 2 h and then rinsed with PBS. This was followed by the deposition of 100 μl RBCs in PBS (106 cells per ml) which was incubated for 10 min after which the cells were shear washed with 60-ml 5P8-10 buffer (5 mM Na$_2$HPO$_4$/NaH$_2$PO$_4$, 10 mM NaCl, pH 8.0) using a syringe at a ~20° angle from the surface plane. The cytoskeleton-exposed samples were dried with nitrogen flow and kept in a desiccator overnight (or longer).

**AFM cell stiffness measurements.** RBCs floated in RPMI medium without Albumax were deposited onto freshly cleaved mica for 5 min and then washed with the same medium by infinite dilution. AFM imaging and indentations were carried out using a JPK Nanowizard III AFM (Berlin, Germany) in QI mode Control Software v.6.1.159. This mode allows the collection of force–distance curves at each pixel, which can be used to acquire topographic images simultaneously with mechanical data. With aid of these images, force curves for analysis were taken near the center of the cell. The elastic modulus was calculated after calibration of the cantilever sensitivity and spring constant, by applying a contact mechanical Hertzian model (using JPK data processing software version 6.1.86) presuming a Poisson ratio of 0.5 and the radius of the colloidal probe tip used. This analysis treats the RBC as a uniform three-dimensional elastic medium, and is used extensively in the literature to characterize changes in the stiffness of the RBC[94–97].

The indentation measurements were conducted with various tips; a gold colloid (R-1.8 μm, spring constant ≈ 0.11 N/m) (sQUBE Co. CP-PNPS-Au-B), a borosilicate glass colloid (R-2.9 μm, spring constant ≈ 0.013 N/m) (sQUBE Co. CP-qp-SCONT-BSG-A)) for which the colloidal probe tip radius was determined by scanning electron microscope (JEOL JSM-7000F) images. We also used a qp-BioAC- Cl CB3 probe (Nanosensors). These probes have a rounded tip apex with typical radius 30 nm, spring constant ≈ 0.06 N/m. For the 200 nm indentation depth which we used with these probes, the tip shape was chosen as a cone with an opening half-angle of 22 degrees in the Hertz model calculations. In all indentation experiments, the cell topography was inspected before and after the indentation to ascertain their integrity after the indentation.

For bortezomib-treated EVs: culture medium containing EVs was treated with 0.05 μM bortezomib, at 37 °C for 2 h, followed by wash via ultracentrifugation. EVs with or without bortezomib were introduced to naïve RBCs for 2 h at 37 °C in albumax-free media before AFM stiffness measurements. For epoxomicin treated-EVs - epoxomicin (HY-13821, MedChemExpress) was used at a concentration of 10 μM. Incubation with RBCs in albumax-free media was done for 2 h at 37 °C before the AFM stiffness measurement.

**AFM cytoskeleton imaging.** Initial AFM imaging was carried out by using a MultiMode AFM with Nanoscope V electronics (Bruker AXS SAS, Santa Barbara, CA) controlled with Nanoscope 9.2 software (Build R2Sr1.130547). Scans were made in ac mode using a silicon probe (Olympus Co. AC240) or in PeakForce Tapping mode using a Super Sharp Silicon tip on a silicon nitride lever (Bruker Co. SAA-HPI-SS).

The large amount of images necessary to train and test the CNN model required high image throughput. This was achieved with a fast-scan AFM system, using photothermal off-resonance tapping and small cantilevers, which fits on the base of the commercial MultiMode AFM system. The fast scanning AFM head and electronics were designed by Georg E. Fantner's group, Ecole Polytechnique Fédérale de Lausanne, Lausanne, Switzerland[98], images were collected with custom software from the group of Georg Fantner, EPFL, (IbniAFMController-BetaVersionv2.0.16-20190117 written under Labview environment). Scans were made with a silicon tip on a silicon nitride cantilever- (Bruker FASTSCAN-B). All cytoskeleton images were analyzed using Gwyddion 2.55 software[99].

**Image processing and machine learning.** We applied deep learning[100,101] to classify RBCs as being healthy (non-treated control uRBCs) or infected (*Pf*-derived EVs treated) by evaluation of fast-scanning AFM (fs-AFM) images. Specifically, we applied a convolution neural network (CNN) model that successfully quantified and verified the difference between cytoskeleton images from healthy cells as compared to those which were exposed to *Pf*-derived EVs. Furthermore, the model was applied to cytoskeleton images for which the *Pf*-derived EVs were pre-treated with the bortezomib proteasome inhibitor. The CNN deep learning approach applied here is similar to those applied recently for biomedical and texture image classification[102–110].

Two data sets of images were obtained with the fs-AFM, corresponding to two experiments performed on different days, with different AFM tips. Batch 1 contains 60 healthy cytoskeleton images and 65 *Pf*-derived EV-exposed cytoskeleton images, and batch 2 contains 51 healthy (no EVs) cytoskeleton images and 58 *Pf*-derived EV-exposed cytoskeleton images. The size of each 2 μm × 2 μm image is 512 × 512 pixels. At this scale and resolution, the number of usable images can be increased by dividing each image into four non-overlapping images of size 256 × 256 pixels. This increased the data set size by a factor of four to 500 images in batch 1 and 436 images in batch 2.

Five different CNN architectures were then examined. For tuning the different CNN architectures, 20% of the images were taken as a test set (with stratification), i.e., 100 and 88 images for testing in batches 1 and 2 respectively. The remaining 80% 512 × 512 images were further randomly grouped (with stratification) into 20 different sets, 75% for training and 25% for validation. Again, each image in each set was divided into four non-overlapping images of size 256 × 256 pixels.

In training/validation splits, the training set alone was image-augmented by four rotations: 0°, 90°, 180°, and 270°, and two flipping operations – horizontal and vertical. The image augmentation increased the number of images in the training set to exceed 2000 in both batches. This has been suggested as an approach to reduce over-fitting[101]. This augmented training set was trained until it reached an accuracy of 96% on training or exceeded 30 epochs. To define the discriminative features in each data set, features from the last feature activation maps are weighted according to their importance, and summed to give the class activation map which ranks the importance of various surface features. Then the class activation map is upsampled to the size of the input image. For obtaining the class activation maps we used the trained model with the necessary modification as described in[45] and fine-tuned the weights of the last layer (obtained from the global 2D average pooling) that connected to the output unit.

After training the model, we performed accuracy evaluation on the validation sets.

On average, we found that the CNN architecture presented in Supplementary Fig. S13A gives the highest accuracy on the validation sets compare to the different architectures tested (different number of filters, neurons, and layers), with average accuracy of ~90%. To obtain further insights, we use this architecture and trained the model again on all data (training + validation, except the testing set), with the image augmentation transformations described, to use it for prediction on testing set.

With the model presented in Supplementary Fig. S13A we obtained accuracies of 99.0% and 96.6% on the test set for data sets 1 and 2, respectively. The prediction probability for healthy cytoskeleton images in the test set for data sets 1, 2 is shown in Supplementary Fig. S13B, C, respectively.

It is clear that in both data sets the model successfully differentiates between the two image classes, namely healthy cytoskeleton (RBC, no EV control) cell images vs. *Pf*-derived EV - exposed cytoskeleton (exo) cell images. We note that the images in data sets 1 and 2, correspond to two different experiments. This means that the images relate to two different populations. Hence, to generalize the model for the two data sets together, a new model should be trained on the full set of images from both batches. The accuracy on the test set in that case is 86.1% with a more-complex model architecture than in Supplementary Fig. S13A, which accounts for the complexity of the two data sets together.

In a second set of experiments, we examined the effect of bortezomib-treatment on the *Pf*-derived EVs. The model classifies 94.1% (188 images in total in batch 1) and 86.0% (236 images in total in batch 2) of the images as healthy. The prediction probability for healthy cytoskeleton images for data sets 1 and 2 is shown in Supplementary Fig. S13D, E, respectively, for the *Pf*-derived EVs that were treated with bortezomib. It is clear that the effect of the bortezomib treatment on the EVs leads to significant retention of cell structure with the majority of the images being classified as healthy[26].

All code was written with Python v. 3.6.8. Besides its default packages, Scikit-learn v. 0.21.1[111] and Keras v. 2.2.2[112] packages were utilized for machine learning and deep learning implementations, together with Scipy v. 1.2.1[113] package for data analysis and OpenCV v. 3.4.2[114] package for computer vision.

**Acoustic force spectroscopy cell mechanics experiments**. AFS experiments with RBC are described in detail elsewhere[26]. Briefly, microfluidic chambers from LUMICKS B.V were used, consisting of a monolithic glass chip with a fluid channel inside and a piezoelectric element on top (as described in[25]). RBCs were flushed into the microfluidic chamber, settled and adhered to a poly-l-lysine functionalized surface. Next, concanavalin-A coated polystyrene beads were flushed in, settled down, and attached to RBCs. Upon voltage application, beads were pulled towards the node of the acoustic pressure field in the chamber, thus pull the cells and leading to their elongation. For imaging, an inverted bright-field microscope was used, as described in[115]. Digital camera images were recorded, and the *x*, *y*, and *z* coordinates of all the tracked microspheres were determined in real-time. The *x* and *y* positions were obtained by quadrant interpolation[116] and the *z* position by means of a look-up table[117]. Force calibration was performed by quantifying and averaging the response of freely suspended beads to force, as previously described[26,115]. To quantify the RBCs viscoelastic response, the data was fitted to a Burger's viscoelastic model. Fitting parameters were extracted using a custom-made Python script.

**Statistical analysis**. Unless otherwise stated, comparisons between two treatments were done with *t*-tests. If experiments were run in blocks (i.e. one replicate of each treatment per block), this was compared with paired *t*-tests. For comparisons between more than two groups, we used one-way ANOVA, followed by Tukey's post-hoc test where it was relevant. If a batch effect was required, we used a two-way ANOVA. For analysis of counts, we used a mixed GLM assuming binomial distribution. Analyses were conducted in R, v. 4.0.3 and using Origin 8 and the implement algorithm in the OriginPro 2018 v. b9.5.0.193 and OriginLab Pro 2018b v. 9.55 software.

**Figure preparation**. Graphs were prepared using Graphpad PRISM v.8 and Microsoft Office Excel 2016. Figures were prepared using Adobe Illustrator CS3 v. 13.0.1.

**Reporting summary**. Further information on research design is available in the Nature Research Reporting Summary linked to this article.

## Data availability
The datasets generated during the current study are available from the corresponding author on reasonable request. All phosphoproteomic proteomic data were deposited in PRIDE partner repository as Project PXD018154. Proteomics data have been deposited to the ProteomeXchange Consortium in PRIDE partner repository as Project PXD023353 and 10.6019/PXD023353. Source data are provided with this paper.

## Code availability
The code used for the Acoustic Force Spectroscopy (AFS) cell mechanics experiments can be found in the following link: https://gitlab.com/sorkin.raya/Cellular-mechanics-AFS/tree/master.

The code for image processing and machine learning is composed of many scripts for each sub-task. Relevant scripts will be supplied from the corresponding author by request.

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

## Acknowledgements

We thank the Malaria Research Reference Reagent Resource Center (MR4) for their generous supply of parasite strains. We thank the laboratory of Alan Cowman and the proteomics unit from the Walter and Eliza Institute, Melbourne for the scientific work on EV proteomic analysis. We thank Dr. Adi Naamati from the University of Cambridge for scientific discussions. We thank the laboratory of Prof. Georg Fantner from EPFL Lausanne for design and assistance with implementation of the fast-scanning AFM. We thank Moran Shalev, Ari Elson, and Haim Barr from the Weizmann Institute for assistance with the kinase assays. This research was supported by a Weizmann Institute Staff Scientist Grant for ZP. The research of OA is supported by the David Barton Center for Research on the Chemistry of Life and the Jeanne and Joseph Nissim Center for Life Sciences Research. OA is the incumbent of the Miriam Berman Presidential Development Chair. We thank Timo Betz for useful discussions. The research of N.R.-R. is supported by the Benoziyo Endowment Fund for the Advancement of Science, the Jeanne and Joseph Nissim Foundation for Life Sciences Research and the Samuel M. Soref and Helene K. Soref Foundation. N.R.-R. is supported by a research grant from David E. and Sheri Stone and by a research grant from Richard and Mica Hadar. N.R.-R. is the incumbent of the Enid Barden and Aaron J. Jade President's Development Chair for New Scientists in Memory of Cantor John Y. Jade. N.R.-R. is grateful for the support from the European Research Council (ERC) under the European Union's Horizon 2020 research and innovation program (grant agreement No. 757743), and the Israel Science Foundation (ISF) (619/16 and 2235/16). M.S. is grateful for the support of a Starting Grant from the ERC (Horizon 2020/ERC grant agreement No. 636752), and for an Israel Science Foundation grant (300/17). M.S. is the incumbent of the Aharon and Ephraim Katzir Memorial Professorial Chair. N.S.G. is the incumbent of the Lee and William Abramowitz Professorial Chair of Biophysics.

## Author contributions

E.D, D.Y., M.S., and N.R.R. designed the experiments and analyzed the data. I.R.G., S.R. C., and N.S.G designed and performed the AFM experiments. T.Z., T.N., E.K., M.A.P., and X.S. performed the proteomic experiments and data analysis. M.I.M., O.A., and Y.O. B performed and analyzed the EV fusion assays. Y.O.B, P.A.K., S.M., T.B.T., A.R., O.Y.R., and D.A. assisted in parasite culturing, EV production and biochemical assays. R.R. assisted in the statistics analyses, D.M. performed EM imaging experiments, I.A. established the machine learning approach, Z.P. assisted in data analysis of FACS-derived data, G.B., R.S., and G.J.L.W. designed, performed and analyzed the experimental system of the Acoustic Force Spectroscopy, T.G.C. assisted in data analysis of the phospho-proteomics and in design of kinase tools. G.B.N., M.S., and N.R.R. analyzed the data and wrote the manuscript.

## Competing interests

The authors declare no competing interests.
