## [Peer Review File · Nature Communications]

Reviewers' Comments:

Reviewer #1:

Remarks to the Author:

The authors have submitted a revised version of their manuscript, which addresses some of the questions raised by this reviewer during the initial review process. However, some of the initial concerns still remain and, in addition, new queries have emerged.

initial review

1. The conclusions drawn rely on the assumption that the extracellular vesicles fuse with the red blood cell plasma membrane and deliver their cargo into the erythrocyte cytosol. While this is possible and even likely given the data presented by the authors, it has not yet been formally demonstrated.

The authors now show that EVs can fuse with large unilamellar vesicles. These additional data are convincing. However, it would have been much more supportive of the key hypothesis if the authors had shown EVs fusion with uninfected red blood cells.

2. What is the evidence that the extracellular vesicles are shed from infected erythrocytes and are not formed or released during parasite egress and red blood cell rupture?

The authors have not experimentally addressed this question.

3. Table 1, please provide the names of the proteins, in addition to the UniProt-IDs.

The authors have complied with this recommendation.

4. Several subfigures feature bar charts. Bar charts are problematic in the sense that they obscure the variance in the data set. For this reason, many journals have banned bar charts. The authors should use other forms of graphically displaying the data or overlay the bars with the original data points. The authors have ignored this advice.

New queries:

5. Figure 1A is confusing. Why does the parasitemia of uRBC-derived EVs drop? It is recommended to plot the original parasitemia levels (in %), without normalization. In addition to the parasitemia of cultures pretreated with uRBC derived EVs and Pf-derived EVs, the parasitemia of non-pretreated cultures should be shown.

6. It is unclear what Figure 1D shows. The whole paragraph describing the AFM experiments, including the conclusions, is overly speculative. For instance, the data do not show that "for the health RBC, the cytoskeletal network is composed of spectrin filaments (of length ~80 nm on average)". It is also pure speculation that "the node filament network seems to be dissolved and replaced by a uniform distribution of membrane proteins" or "that the cytoskeletal proteins have aggregated". A stretched or a disrupted membrane skeleton are frequent preparation artefacts of AFM measurements. Have the authors considered this?

7. The apparent specific enrichment of a functional 20S proteasome and various human and *P. falciparum* kinases is surprising and suggests a selection and sorting process. How do the authors explain this observation?

8. *P. falciparum* has evolved a very sophisticated protein sorting and trafficking mechanism to deliver proteins into the red blood cell compartment. This includes targeting motifs, translocons, specialized

proteases to process exported proteins, Maurer's clefts, etc. Now the authors propose that *P. falciparum* proteasome subunits and numerous kinases are somehow released from the extracellular parasite across two membranes, the host cell cytoplasm, into the environment as components of EVs. This reviewer does not see how this would work. It is more likely that EVs form during rupture of infected erythrocytes at the end of the 48 hr replicative cycle as the daughter parasites egress.

9. The Young's modulus of red blood cells depends on the load and the cellular position where the measurement is made - at the center or the periphery. Irrespectively, previous studies report values in the kPa range, whereas the presented study observed values in Pa range. How is this explained? More experimental detail should be provided to clarify these issues.

10. Figure 3D and corresponding legend. Statistical analysis using a paired t-test does not seem to be justified. See also statistical analysis of data presented in 3B and 3C where two-tailed t-tests were used. Judging by the error bar in Fig. 3D, the data are probably not significantly different, using a Student's t-test.

11. Figure 3F. How can the authors exclude that brotezomib inhibits the parasite's proteasome and, in turn, parasite development.

12. Figures 1C/D and 4A/C. Although in both figures no EVs and Pf-derived EVs were investigated, the AFM histograms and images look different. What is the explanation?

13. Figure 6. Again, what are the evidence that the EVs are shed from infected erythrocytes before rupture?

14. The manuscript, including figure legends, would benefit from substantial editing to improve readability.

Reviewer #2:

Remarks to the Author:

The author's did an excellent job of addressing my questions and I have no additional comments or questions.

Reviewer #4:

Remarks to the Author:

Dekel et al. describe in their manuscript a new discovery related to extracellular vesicle (EV) containing 20S proteasomes that are derived from malaria parasites and prime RBCs for further parasite invasion. This increased parasitemia is in part mediated by EV-20S targeting of several cytoskeletal proteins within the host cell which alters membrane stiffness. This study uses a series of technical approaches to specifically identify the nature of these EV-20S proteasomes, their substrates, and the impact on malaria invasion. It is a rigorous and novel study. In a previous review of this manuscript I supplied a series of critiques. At this time all of my critiques and concerns have been satisfied either by further explanation, new experimentation or increased controls. This is a very interesting and rigorous study.

Reviewer #5:

Remarks to the Author:

In this manuscript, Dekel et al describe a novel mechanism by which *P. falciparum* parasites prepare naïve RBCs for invasion. This priming occurs through the secretion of extracellular vesicles containing a proteasome complex and kinases that remodel the cytoskeletal network of naïve RBCs. This process softens the cell and facilitates parasite invasion.

The authors addressed the most important points raised by the reviewer, but some concerns still need to be discussed or addressed in simple experiments.

Although they provide a quantitative assessment of the EV concentration in their experiments (1012 particles per ml), this concentration is much higher than the EV concentration in patient samples (estimated at 4.105 per ml in PMID: 21282195). It is therefore questionable whether the observed changes are physiologically relevant. The authors should discuss this point.

The AFM cytoskeleton imaging is interesting and reveals an EV-mediated expansion of the spectrin meshwork, which is a phenotype already described in RBCs infected with *P. falciparum* (PMID: 25950144, PMID: 27071094, PMID: 12794267). However, in the previous studies this increase in spectrin mesh size correlated with reduced membrane deformability, unlike in the present manuscript. The authors should discuss this point.

RBC cytoskeleton reorganization upon incubation with EVs would be expected to be accompanied by increased fragility of the RBCs. RBCs mechanical cellular fragility can be easily addressed by classical osmotic fragility experiments or by ektacytometry.

Reviewer #1

The authors have submitted a revised version of their manuscript, which addresses some of the questions raised by this reviewer during the initial review process. However, some of the initial concerns still remain and, in addition, new queries have emerged.

1. The conclusions drawn rely on the assumption that the extracellular vesicles fuse with the red blood cell plasma membrane and deliver their cargo into the erythrocyte cytosol. While this is possible and even likely given the data presented by the authors, it has not yet been formally demonstrated.

The authors now show that EVs can fuse with large unilamellar vesicles. These additional data are convincing. However, it would have been much more supportive of the key hypothesis if the authors had shown EVs fusion with uninfected red blood cells.

We thank the reviewer for indicating that the additional data are convincing. Demonstrating EV fusion with uninfected red blood cells will indeed support our findings that the EV fuse and deliver their cargo to the red blood cells, we believe that adding it is beyond the scope of this work. The Editor indicated that this point can be considered as such.

2. What is the evidence that the extracellular vesicles are shed from infected erythrocytes and are not formed or released during parasite egress and red blood cell rupture?

The authors have not experimentally addressed this question.

We would like to emphasize that the egress stage is not relevant to this manuscript.

To avoid any egress-related vesicle structures all our EV assays were from highly synchronized parasite culture, 24 hours post invasion into the RBC, at the trophozoite stage, and long before their egress.

We added the clarification to the materials and methods.

It is known that EVs are actively produced by most cell types (PMID: 29339798; PMCID: PMC7379748; PMCID: PMCID 5318253; PMID: 25536932; PMCID: PMC3575529; PMID: 23584393; PMID: 30792291; PMID: 24566916; PMID: 25488940; PMID: 25704309).

3. Table 1, please provide the names of the proteins, in addition to the UniProt-IDs. The authors have complied with this recommendation.

We thank the reviewer for this positive comment.

4. Several subfigures feature bar charts. Bar charts are problematic in the sense that they obscure the variance in the data set. For this reason, many journals have banned bar charts. The authors should use other forms of graphically displaying the data or overlay the bars with the original data points.

The authors have ignored this advice.

Wherever possible we replaced the bar charts. In Figures 1 and 4 we have provided box plots with explanatory text. It is true that in addition, the bar charts were left because we think that they provide a simple, graphic view of the data.

Nevertheless, in response to the reviewer's remark we added **New Supplementary Figures 2, 7 and 14** that include all the experimental raw data. In addition, **New Supplementary Figure 3**, which displays the mechanical cellular fragility of naïve RBC treated with *Pf*-derived EVs, is depicted using individual data points to directly show the spread of the data.

New queries:

5. Figure 1A is confusing. Why does the parasitemia of uRBC-derived EVs drop? It is recommended to plot the original parasitemia levels (in %), without normalization. In addition to the parasitemia of cultures pretreated with uRBC derived EVs and Pf-derived EVs, the parasitemia of non-pretreated cultures should be shown.

We followed the reviewer's suggestion and have now added new Supplementary Figures (**New Supplementary Figures 2 and 7**), that provide the original parasitemia data from all biological replicates and of the three samples received from uRBC-derived EVs, *Pf*-derived EVs and untreated *Pf*.

We decided to retain the parasitemia presentation (within the main figures) as is to be consistent with similar studies in the field, already normalized to non-treated cultures; as shown in various publications (PMID: 28226242; PMID: 21709259; PMID: 20003396; PMID: 32782246; PMID: 22901539; PMID: 4; PMID: 29378713; PMID: 15259022; PMID: 23717205).

6. It is unclear what Figure 1D shows. The whole paragraph describing the AFM experiments, including the conclusions, is overly speculative. For instance, the data do not show that “for the healthy RBC, the cytoskeletal network is composed of spectrin filaments (of length ~80 nm on average)”. It is also pure speculation that “the node filament network seems to be dissolved and replaced by a uniform distribution of membrane proteins” or “that the cytoskeletal proteins have aggregated”. A stretched or a disrupted membrane skeleton are frequent preparation artefacts of AFM measurements. Have the authors considered this?

Figure 1D displays very important data that visually show the degradation of the cytoskeleton. A clear difference is seen between the upper and lower images. The referee suggests that the differences we see are the result of preparation artifacts. We strongly disagree with this comment as differences between control and EV-treated samples should not be due to preparation since all samples underwent the same procedure, which has been well documented in the literature, (see, for instance, Sinha *et al*, PMID: 25950144 and Dearnly *et al*, PMID: 27071094). Furthermore, our images of healthy cells are very similar to what is seen by others, as we have written. These differences are striking and repeatable. One could make the claim that the preparation procedure has a stronger effect on the damaged cells than on the healthy ones, but in the unlikely event that this is true it would be further evidence that changes occurred with the EV-exposed cells. These images furthermore went an extensive CNN analysis as described in the paper and SI to remove any subjectivity from their interpretation and the results of that analysis are very convincing in their support of our claims.

The referee also terms our discussion speculative. No one measurement in this complex system can provide a definitive conclusion, but must be considered together with the full set of our results (for instance, the biochemical assays showing the fate of cytoskeletal proteins). The relation between structural changes and distribution of membrane proteins is an example of this interplay. The 80 nm length was referenced as a descriptor for the nature of the network which is known from the literature, to which we compare our results and intended only to indicate that our results produce cytoskeleton images similar to those shown by others for the healthy cell. We remove now the mention of 80 nm length and made other minor modifications in the text to keep the wording in this results section more observational, reserving explanations for discussion. However, we do not feel that our conclusions should be changed nor do we feel that they are speculative considering the varied, independent approaches they derive from.

7. The apparent specific enrichment of a functional 20S proteasome and various human and *P. falciparum* kinases is surprising and suggests a selection and sorting process. How do the authors explain this observation?

This is a valid and important question, and we are also curious about the sorting process that leads to enrichment of functional 20S proteasome and various human and *P. falciparum* kinases, but feel that it falls outside the scope of this study. Any explanation provided at this time will be speculative.

8. *P. falciparum* has evolved a very sophisticated protein sorting and trafficking mechanism to deliver proteins into the red blood cell compartment. This includes targeting motifs, translocons, specialized proteases to process exported proteins, Maurer's clefts, etc. Now the authors propose that *P. falciparum* proteasome subunits and numerous kinases are somehow released from the extracellular parasite across two membranes, the host cell cytoplasm, into the environment as components of EVs. This reviewer does not see how this would work. It is more likely that EVs form during rupture of infected erythrocytes at the end of the 48 hr replicative cycle as the daughter parasites egress.

While we appreciate the reviewer's feedback, we respectfully disagree, as it is well documented that organisms produce EVs, including the malaria-infected RBCs (more than 80 papers on EVs in malaria research).

In respect to the exact mechanisms of EV biogenesis, cargo loading and sorting, while these are very interesting and important directions of investigation they are out of the scope of this paper.

9. The Young's modulus of red blood cells depends on the load and the cellular position where the measurement is made - at the center or the periphery. Irrespectively, previous studies report values in the kPa range, whereas the presented study observed values in Pa range. How is this explained? More experimental detail should be provided to clarify these issues.

As stated in Fig. 1 legend, all measurements were made near the cell center. We now add this point also to the experimental section, as well as a statement regarding our sample preparation method:

Previous works which the author refers to bound the cells to substrate with glutaraldehyde. This leads to significant stiffening of the cells as documented in the literature. See for instance (S. Barns, et al, PMID: 29258590) G. Tomaiuolo PMID: 25332724) I. Dulińska-Molak, et al PMID: 24489527.

Works which take care to use mild bonding, as we have, report similar values to what we observe: M. Li, et al. PMID 23160828.

Notably, we have now added new mechanical data using a supplementary technique of Acoustic Force Spectroscopy to support the AFM results (**New Supplementary Figures 3A and 3B**).

10. Figure 3D and corresponding legend. Statistical analysis using a paired t-test does not seem to be justified. See also statistical analysis of data presented in 3B and 3C where two-tailed t-tests were used. Judging by the error bar in Fig. 3D, the data are probably not significantly different, using a Students t-test.

The reviewer is correct, in Figure 3C and 3D statistical analysis was done using a paired t-test. The statistical analyses were competently performed by Dr. Ron Rotkopf, a qualified biostatistician. Specifically, a paired t-test was used, since the experiments were conducted in blocks (or batches) – each block contained one replicate of each treatment. The differences between blocks are sometimes larger than the main effect we are testing, but the directionality of the differences is consistent within each block (as can be seen in the figure below), therefore we believe that this is a reliable result. Nevertheless, in response to the reviewer’s remark we added **New Supplementary Figures 2, 7 and 14** that include the experimental raw data.

The figure shows a graphic representation of the data used to generate the graphs in Figures 3C and 3D. It demonstrates the consistent directionality of the differences between uRBC-derived EVs and Pf-derived EVs in each experiment (batch).

11. Figure 3F. How can the authors exclude that brotezomib inhibits the parasite’s proteasome and, in turn, parasite development.

We have edited the text to clarify that EV treated with brotezomib were washed with PBS, followed by ultracentrifugation spin prior to their introduction to naïve RBCs.

12. Figures 1C/D and 4A/C. Although in both figures no EVs and Pf-derived EVs were investigated, the AFM histograms and images look different. What is the explanation?

As in any biological sample, variability can occur, and the preparation process itself can introduce additional variability. In any event, we do claim that our images can be compared favorably with the previously published AFM cytoskeleton images. Note, for instance, the significant number of “holes” in the middle panel image of control, shown in the work of Sinha *et al*, PMID: 25950144 their Fig. 2, and the similarity of our images to those of Liu *et al* PMID: 12794267 Figs. 4 and 5). In the highest resolution images, our Fig. 4C, the “holes” are of the size and appearance of the holes that have been observed in the spectrin filament network (see references above, and PMID: 15866743).

We would like to stress that we have included these images in order to give a visual guide to the damage done to the cytoskeleton network but due to natural variability and to avoid subjective assessment, we performed the extensive CNN analysis provided in the text which proves this difference objectively and unequivocally (**Figure 4E and Supplementary Figure 13**).

We would like to draw attention to the fact that the box plots for the no EV samples in both figures show very similar statistics. As described in figure captions, the histogram representation between the two figures cannot be directly compared, Fig. 1 is an average of all measurements on each cell and Fig. 4 shows all the individual measurements.

13. Figure 6. Again, what are the evidence that the EVs are shed from infected erythrocytes before rupture?

We have used highly synchronized parasite culture, 24 hours post invasion, at the trophozoite stage and long before the egress phase. In the revise manuscript, we clarify this point.

14. The manuscript, including figure legends, would benefit from substantial editing to improve readability.

After revising our manuscript to address the reviewer’s comments, we have had it rechecked by a professional English editor. Consequently, grammatical and stylistic edits have been made throughout the text. We hope that this revised manuscript meets the reviewer’s expectations.

Reviewer #5

In this manuscript, Dekel et al describe a novel mechanism by which *P. falciparum* parasites prepare naïve RBCs for invasion. This priming occurs through the secretion of extracellular vesicles containing a proteasome complex and kinases that remodel the cytoskeletal network of naïve RBCs. This process softens the cell and facilitates parasite invasion.

1. The authors addressed the most important points raised by the reviewer, but some concerns still need to be discussed or addressed in simple experiments.

We thank the reviewer for his insightful comments and for indicating that the most important points were already addressed.

2. Although they provide a quantitative assessment of the EV concentration in their experiments (1012 particles per ml), this concentration is much higher than the EV concentration in patient samples (estimated at 4.105 per ml in PMID: 21282195). It is

therefore questionable whether the observed changes are physiologically relevant. The authors should discuss this point.

We understand the reviewer's concern. The reviewer is referring to a paper from 2011. During the past ~10- years the EV field has developed rapidly with better detection technologies (as Nanosight Tracking Analysis, NTA), providing higher resolution and accuracy to detect also the smaller nano-vesicles; it became clear that the EV sizes are ranging from ~30-300 nm, thus a much larger population. In PMID: 21282195 they could not detect these populations due to the limitation of their detection method. Thus, the NTA method (used by us) is the most common and accepted technique to monitor EV concentrations to date.

Indeed, using NTA measurements of clinical samples derived from malaria patients, several recent studies demonstrated much higher concentrations of EVs, then that reported by Nantakomol *et al.*, and in agreement with our report.

For instance two report from this year:

1. PMID: 32487994: describes the NTA as a mean to determine EV concentration, that yielded 4×10^9 per ml.
2. PMID: 32002165 detected concertation within the range of $1.5-1.83 \times 10^{10}$ per ml, again by NTA measurement.

Therefore, making our observation in the physiological range. We have revised the manuscript to include citations of these two recent papers.

3. The AFM cytoskeleton imaging is interesting and reveals an EV-mediated expansion of the spectrin meshwork, which is a phenotype already described in RBCs infected with *P. falciparum* (PMID: 25950144, PMID: 27071094, PMID: 12794267). However, in the previous studies this increase in spectrin mesh size correlated with reduced membrane deformability, unlike in the present manuscript. The authors should discuss this point.

We thank the reviewer for his comment and would like to initially respond specifically to the reviewer's more general question, as to whether we can relate a specific mesh size to RBC deformability. To answer this, we relate to the specific works mentioned by the referee, indicating why they may be expected to yield a different result than ours.

Reference 7 in our work, Sinha *et al* (PMID: 25950144), examined the effect of oxidative stress on RBC deformability (not interaction with EVs or malaria parasites). Indeed, three out of 4 oxidants tested increased the stiffness, whereas only one reduced it. However, they also point to other oxidants, not tested in their work, that lead to improved cell fluidity and deformability.

They go on to compare their work with three other studies that indeed show that oxidation leads to decrease in stiffness and note:

“Our results, therefore, do not necessarily contradict the previous findings, but should rather be seen as complementary sets of information using an alternate technique that measures deformability of the outer membrane shell consisting of membrane and cytoskeleton. It also highlights the fact that there are different ways to measure RBC deformability focusing on different aspects (or combination of different aspects), and we would need to be careful in comparing the results obtained using different experimental methods”.

We therefore view these results as supportive of the trend that we found, namely that softer RBCs permit enhanced malaria invasion, while rigid RBCs inhibit it.

The focus of our study was the preliminary stage of cell “conditioning” before actual infection. In a study on RBCs directly exposed to the *Pf* parasite, Suresh *et al.* (PMID: 16701777) see a continual stiffening of the RBC beginning with initial exposure and continually developing as the cell undergoes further stages of infection. Therefore, *we avoided studying infected RBCs*. In the present study we exclusively test the effect of purified secreted EVs and removed excess material which exists in condition medium of parasite culture.

Regarding (PMID: 25950144, PMID: 27071094, PMID: 12794267), none of these addressed the effect of *Pf*-derived EVs on the cytoskeleton healthy cells, as summarized here:

PMID: 25950144 (discussed above).

PMID: 27071094 This study investigated the cytoskeleton at different stages of the parasite growth. And showed it is correlated with reduced membrane deformability. As explained above, our measurements are relevant to the early stages of invasion (pre-infection) and are different than the later stages investigated in this referenced work.

PMID: 12794267 did not investigate any effect related to *Pf* on the cytoskeleton but rather aging. Furthermore, this work provides images, but no mechanical measurements. While, the aged cells appearance was different from ours, displaying a dense cytoskeleton network, the healthy cell cytoskeleton images are similar to ours.

Several studies have shown that *P. falciparum* invasion initiates with a drop in the elastic modulus of the RBC. For example, Sisquella *et al* (our ref. 6 PMID28226242) show that recombinant proteins lower the RBC elastic modulus upon binding. This process is accompanied by phosphorylation of the cytoskeleton, supporting our data.

In summary, we conjecture that the differences and similarities of our study to those of others are related to: (1) Different mechanical properties revealed by the different techniques, and (2) the specific nature of the biochemical process being studied, and stage of the infection process.

4. RBC cytoskeleton reorganization upon incubation with EVs would be expected to be accompanied by increased fragility of the RBCs. RBCs mechanical cellular fragility can be easily addressed by classical osmotic fragility experiments or by ektacytometry.

This is a valid comment raised by the reviewer. We have now added new mechanical data using a supplementary technique of Acoustic Force Spectroscopy to support the AFM results.

Mechanical cellular fragility is a clinical test, performed on a bulk sample to measure cell rupture under mechanical stress. Cell rupture and deformation are two different phenomena and the former has interesting mechanical implications for the functioning of the RBC as they traverse the body, but does not seem to have direct relevance to the early stage of cellular invasion by the malaria parasite, nor the initial stages preceding that which we have studied and therefore was not investigated in this study. We propose that the remodeling we observe is the first stage of invasion. Only in much later stages, does the cell stiffen and possibly rupture.

We do agree that such additional, independent mechanical measurements can strengthen our claims. Consequently, in the revised manuscript we provide additional data using Acoustic Force Spectroscopy measurements, **New Supplementary Figures 3A and 3B**.

Using this technique, an acoustic pressure field was used to apply well-controlled forces up to 500 pN to cells confined between a microsphere and a surface. The microspheres which are attached on top of cells are pushed towards the nodes of the acoustic pressure field, thereby pulling on the cells. The three dimensional position of the beads was tracked using a standard optical microscope, which monitored the elongation of cells under the applied force. Using this approach, we found that RBCs treated with *Pf*-derived EVs elongated significantly more than non-treated RBCs. Such experiments allow quantitative determination of the stiffness, which show, as for the AFM measurements, greater compliance for EV- treated RBCs.

Reviewers' Comments:

Reviewer #1:

Remarks to the Author:

The authors have substantially improved the manuscript both in terms of supporting experimental evidences and readability of the text. The results are now plausible and they appear to be robust. The reviewer also agrees with the authors that additional data would be beyond the scope of this article. Having said this, the reviewer is still worried about some of the conclusions drawn. However, it is the prerogative of the author to interpret the finding in light of a certain hypothesis. Time will tell whether the favored hypothesis is right.

Apart from that, it is a very original, provocative and possibly ground breaking study that deserves to be aired. Please accept my apologies for my initial criticism - which was meant to help improve the study.

Reviewer #5:

Remarks to the Author:

The authors adequately addressed my queries. I have no additional comments on this manuscript.